

# Exploring phytochemicals as potential pharmacological inhibitors for NS1 protein of Kyasanur forest disease virus using virtual screening, molecular docking, and molecular simulation approach

Sharanappa Achappa[1], Nayef Abdulaziz Aldabaan[2], Ibrahim Ahmed Shaikh[2], Mater H. Mahnashi[3], Shivalingsarj V. Desai[1], Mufarreh Asmari[3], Mohammed Alasmary[4], Uday M. Muddapur[1], Basheerahmed Abdulaziz Mannasaheb[5] and Aejaz Abdullatif Khan[6]

[1] Department of Biotechnology, KLE Technological University, Hubballi, Karnataka, India
[2] Department of Pharmacology, College of Pharmacy, Najran University, Najran, Saudi Arabia
[3] Department of Pharmaceutical Chemistry, King Khalid University, Abha, Saudi Arabia
[4] Department of Medicine, College of Medicine, Najran University, Najran, Saudi Arabia
[5] Department of Pharmacy Practice, College of Pharmacy, AlMaarefa University, Riyadh, Saudi Arabia
[6] Department of General Science, Ibn Sina National College for Medical Studies, Jeddah, Saudi Arabia

Corresponding authors
Shivalingsarj V. Desai,
desaisv@kletech.ac.in
Mohammed
Alasmary, myalasmary@nu.edu.sa

## ABSTRACT

**Background**. Kyasanur forest disease virus (KFDV) remains a significant public health challenge due to the limitations of existing vaccines, creating a critical need for effective antiviral treatments. KFDV is a tick-borne virus responsible for 400–500 new cases annually, with a mortality rate of 3–5%. The nonstructural protein 1 (NS1), which plays crucial roles in host cell interactions, immune evasion, and viral replication, represents a promising target for antiviral drug development.

**Objective**. This study aims to identify potential antiviral compounds that inhibit the activity of KFDV NS1 protein using a computational pharmacological drug design approach. The objectives include determining the 3D structure of the NS1 protein through homology modeling, conducting virtual screening of phytochemicals to identify potential inhibitors, and performing molecular dynamics simulations to assess the stability and binding free energies of the selected compounds.

**Methods**. The 3D structure of KFDV NS1 protein was predicted using homology modeling and validated using Ramachandran plot analysis. Virtual screening of phytochemicals from the Indian Medicinal Plants, Phytochemistry And Therapeutics (IMPPAT) database was performed to identify potential NS1 inhibitors. The top 15 compounds with the highest binding affinities were selected and subjected to absorption, distribution, metabolism, excretion, and toxicity (ADMET) analysis. Molecular dynamics simulations were conducted in duplicates for 200 ns to evaluate the stability of the ligand-NS1 complexes, and an additional independent simulation with randomized initial velocities was performed to ensure statistical robustness. Binding free energies were calculated using the molecular mechanics-generalized born surface area (MM-GBSA) method to determine the binding strength of each compound.
**Results**. The 3D structure of the KFDV NS1 protein was determined using I-Tasser-MTD, Robetta, and Swiss Model servers, and a minimized model of I-Tasser, achieving an ERRAT score of 94.37, was selected. Virtual screening of 11,530 phytochemicals from the IMPPAT database identified the top 115 compounds after three screening phases. Out of the 15 screened compounds, L2, L3, and L5 demonstrated notable binding affinities of −9.34, −9.12, and −9.08 kcal/mol, respectively, compared to the FDA-approved antiviral dasabuvir, which had a binding affinity of −8.0 kcal/mol. Molecular dynamics simulations confirmed the stability of compounds L2 (IMPHY010294), L3 (IMPHY001281), L5 (IMPHY011162), and dasabuvir, with free-energy binding values of −62.97 ± 4.0, −77.22 ± 4.71, −62.07 ± 2.88, and −87.68 ± 4.31 kcal/mol, respectively.

**Conclusion**. The computational analysis suggests that compounds L2 and L3 have strong binding affinities comparable to dasabuvir, indicating their potential as pharmacological inhibitors of the KFDV NS1 protein. Further validation through in vitro assays would complement these *in silico* findings.

## INTRODUCTION

Kyasanur forest disease (KFD), also known as monkey fever, is a tick-borne viral infection that has been endemic to the western coast of India since its identification in 1957. Despite its long history, KFD has increasingly spread to new regions (*Achappa et al., 2024*; *Chakraborty et al., 2024*) and remains insufficiently addressed in terms of research and public health response (*Purse et al., 2020*; *Holbrook, 2012*). The Kyasanur forest disease virus (KFDV), responsible for this disease, is transmitted to humans primarily through the bites of ticks from the species *Haemaphysalis spinigera* and *Haemaphysalis turturis* (*Srikanth et al., 2024*). KFDV is classified within the Flaviviridae family and is related to other members of the Russian spring-summer encephalitis virus group, including the Alkhumra virus and the Omsk hemorrhagic fever virus (*Holbrook, 2012*). Historically confined to certain districts in Karnataka, KFD has progressively spread to neighboring areas such as Chamarajanagar, Chikmagaluru, Udupi, North Kannada, and South Kannada (*Kuchelkar et al., 2024*). Recent outbreaks have also been reported in Kerala, Tamil Nadu, Maharashtra, and Goa, indicating a broader geographical distribution of the disease (*Srilekha et al., 2024*). The increasing incidence of KFD in Karnataka, particularly in districts like Uttara Kannada, Shivamogga, and Chikkamagaluru, has raised significant public health concerns. As of February 2024, the surge in KFD cases in Uttara Kannada district, with 49 positive cases and two fatalities, underscores the urgent need for enhanced public health measures (*Verma et al., 2024*).

Clinically, KFD has a 2–7 day incubation period following bites of a tick, leading to high-grade fever that lasts 5–12 days, accompanied by chills, severe headache, vomiting, and diarrhea (*Verma et al., 2024*; *Chunduru & Saravu, 2023*). Hemorrhagic manifestations such as epistaxis, gum bleeding, and intestinal bleeding may occur, though they are observed

in approximately 10% of cases (*Chunduru & Saravu, 2023*). The disease is in its second phase, after a fever-free interval of 1 to 2 weeks; patients experience prolonged fever, headaches, and central nervous system symptoms, including behavioral disturbances, tremors, giddiness, neck stiffness, and abnormal reflexes (*Sebastian, Varma & Gupta, 2024*; *Chandran et al., 2016*; *Gladson et al., 2021*). The pathogenesis of KFDV involves the virus attacking the host's immune response and replicating in various tissues, including the spleen, liver, and lymph nodes, thereby facilitating systemic infection. The immune response of the host has a dual role: it can aid in viral clearance in mild cases while also contributing to immunopathological damage in severe cases (*Shah et al., 2018*).

KFDV is a spherical enveloped virus (diameter of 40–60 nm) with a single-stranded positive-sense RNA genome. The KFDV genome encodes a polyprotein that undergoes proteolytic cleavage to generate three structural proteins (capsid-C, transmembrane-M, and envelope-E) and seven non-structural (NS) proteins (NS1, NS2A, NS2B, NS3, NS4A, NS4B, and NS5) (*Lindenbach & Rice, 2003*; *Sarika et al., 2024*). The E glycoprotein is critical for mediating viral attachment and entry into host cells, while the non-structural proteins play roles in viral replication and immune evasion (*Achappa et al., 2024*). Out of these proteins, the NS1 protein is multifunctional, contributing to viral replication, immune evasion, and pathogenesis (*Rastogi, Sharma & Singh, 2016*). Intracellularly, NS1 forms dimers and hexamers within the endoplasmic reticulum and partners with NS4A to assemble the viral replication complex, creating a protected environment for RNA synthesis and evading host immune detection (*Huang et al., 2024*). Secreted NS1 further facilitates immune evasion by interacting with complement system components such as C4b and C1s, thereby inhibiting activation of the complement system and preventing the formation of the membrane attack complex (MAC) (*Jayaraman, Walachowski & Bosmann, 2024*). Additionally, NS1 disrupts the endothelial cell barrier by binding to glycocalyx and tight junction proteins, leading to increased vascular permeability and contributing to the hemorrhagic symptoms observed in KFD (*Henrio Marcellin & Huang, 2024*; *Dayarathna et al., 2024*; *Beatty et al., 2015*). Given the essential role of NS1 in replication, immune evasion, and pathogenesis, it represents a key target for therapeutic intervention (*Chaudhuri et al., 2024*) for the development of antiviral drugs and vaccines.

Current preventive strategies, including the formalin-inactivated whole virus vaccine, have demonstrated limited effectiveness (*Bhatia et al., 2023*; *Srikanth et al., 2023*), highlighting the need for more effective approaches. The increasing incidence of KFD and its expanding endemic regions further underscores the necessity for improved public health responses and the development of novel therapeutic strategies (*Hafeez et al., 2023*). In response to this need, our research aims to incorporate the identification of potential antiviral agents from the Indian Medicinal Plants Phytochemicals and Therapeutic (IMPPAT) database. The IMPPAT database is a comprehensive repository of plant-derived compounds with documented antiviral activities, derived from traditional medicinal plants used in India (*Vivek-Ananth et al., 2023*). It provides detailed information on the chemical structures, pharmacological properties, and bioactivity of these compounds. By integrating the structural insights obtained from our *in silico* analysis of the KFDV NS1 protein with the data from the IMPPAT database, we aim to identify promising antiviral drug candidates

by an *in silico* method. The *in silico* method of drug discovery provides a way for researchers to accelerate the process of drug discovery, which otherwise is a time-consuming process by the traditional approach (*Oselusi et al., 2024*).

The *in silico* drug discovery process involves understanding the structural and functional characterization of protein (*Hosen et al., 2024*), screening of vast chemical libraries from different drug databases by virtual screening (*Alshehri, Wahab & Almoyad, 2023*), studying the interaction analysis between the drug and the protein by molecular docking (*Sahu et al., 2024*), and understanding these interactions over a time scale and its stability study by molecular dynamics (MD) simulation (*Singh et al., 2024*). Combining these *in silico* techniques increases the ability to find effective drugs for pathogens, thereby reducing the cost, time, and extensive usage of laboratory chemicals and equipment (*Dalbanjan & Praveen Kumar, 2024*).

Recent advancements in drug and vaccine design against Kyasanur Forest Disease Virus (KFDV) have increasingly focused on computational approaches targeting key viral proteins, including the envelope (E) protein, the NS2B/NS3 protease complex, and the NS5 polymerase protein, which are vital to KFDV pathogenesis. The E protein, critical for virus-host interactions, is a major target for vaccine and drug development. *Arumugam & Varamballi (2021)* utilized immunoinformatics to design a multi-epitope subunit vaccine targeting conserved regions of the E protein, predicting antigenic epitopes with potential cross-protection. *Dey et al. (2024)* expanded on this by using molecular dynamics simulations to characterize E protein interactions with host receptors and identify specific epitopes for vaccine development. *Achappa et al. (2024)* employed computational techniques to screen potential inhibitors of the E protein, identifying several promising candidates. The NS2B/NS3 protease complex, essential for viral replication, has also been a focus for drug design. *Kandagalla, Kumbar & Novak (2023)* employed molecular dynamics simulations to model the NS2B/NS3 complex, highlighting allosteric sites for potential inhibitors. Similarly, *Kandagalla et al. (2022)* identified FDA-approved drugs and a natural compound as potential inhibitors of the NS3 protease of KFDV. More recently, in 2024, computational studies continued to explore inhibitors targeting this complex, utilizing ensemble docking approaches to identify compounds that may disrupt viral replication. Another important drug target is the NS5 protein, which plays a key role in viral genome replication and capping. *In silico* studies have focused on identifying inhibitors that target the RNA-dependent RNA polymerase (RdRp) and its interaction with viral RNA. Given the structural similarities between KFDV NS5 and other flaviviral polymerases, drug repurposing efforts have identified existing NS5 inhibitors from related viruses, such as dengue and Zika, as potential candidates for KFDV treatment (*Zong et al., 2025*). These computational studies provide insights into rational drug design, underscoring the significance of *in silico* methodologies in accelerating the development of interventions for KFDV.

By leveraging the advantages of the *in silico* approach in drug discovery, our research aims to determine the 3D structure of the NS1 protein, identify its active site, and discovering the effective antiviral compounds through virtual screening of the IMPPAT database. Molecular docking studies were conducted to understand the interactions between potential drugs
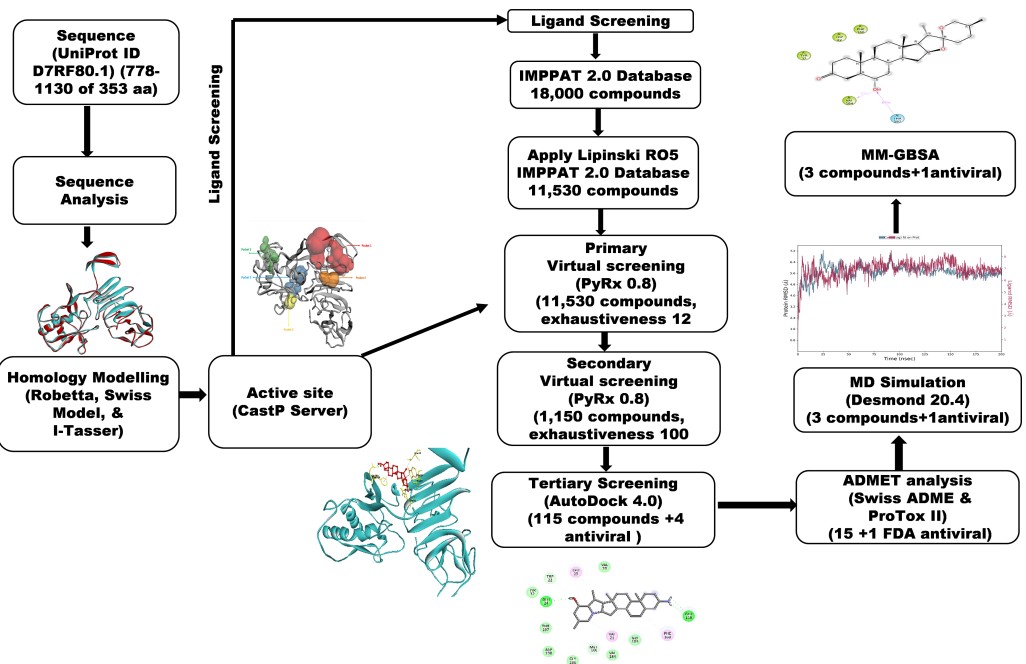

Figure 1 Methodology schema employed in the study for predicting inhibitors for the NS1 protein of KFDV.

and the NS1 protein, followed by molecular dynamics simulations to assess their stability and efficacy. This study aims to provide foundational knowledge for the development of effective *in silico*-based antiviral therapies for KFDV.

## METHODOLOGY

A workflow schema detailing the stepwise techniques employed in this study is shown in Fig. 1.

### Sequence analysis

We used the UniProt database for the extraction of the NS1 protein sequence in FASTA format (*UniProt, 2023*). The ProtParam tool available on the ExPASy website was utilized to investigate the physicochemical characteristics of the NS1 protein under investigation (*Gasteiger et al., 2005*). The BLASTp tool was used for searching homologous sequences using PDB as a database using the BLASTp algorithm (*Johnson et al., 2008*). The NS1 protein's secondary structure analysis was analyzed using SOPMA by keeping default parameters like output width as 70, conformational states as 4, including helix, coil, sheet and turn, window width as 17 and similarity threshold as 8 (*Geourjon & Deleage, 1995*), while it's transmembrane region was characterized using TMHMM 2.0 (*Krogh et al., 2001*).

### 3D structure determination and refinement

We utilized the I-Tasser MTD, Swiss Model, and Robetta systems to predict the 3D structure of the NS1 protein in the absence of a crystallographic framework. I-Tasser

MTD is a hierarchical protocol to predict structures and functions of multi-domain (MTD) proteins (*Zhou et al., 2022*). The Robetta server employs domain division and utilizes either *de-novo* structure prediction or comparative modeling to generate structural models (*Kim, Chivian & Baker, 2004*). The Swiss Model is a web-based application that integrates protein structure homology modeling. It consists of four steps: identifying structural templates, aligning the target sequence with the template structures, creating the model, and evaluating the model's quality (*Waterhouse et al., 2018*). We utilized the energy minimization program YASARA to determine the conformation with the least energy for the 3D structure of the NS1 protein (*Krieger et al., 2009*). After the minimization of the 3D structure, secondary structural analysis was evaluated using the Stride web interface (*Frishman & Argos, 1995*).

## Validation of 3D structure

Different methods for validation of determined 3D structure of NS1 protein was used like ERRAT (*Colovos & Yeates, 1993*) for determining the quality of structure, PROCHECK (*Laskowski et al., 1993*) for stereochemical evaluation for generating Ramachandran plot, QMEAN score (*Benkert, Biasini & Schwede, 2011*) and ProSAweb (*Wiederstein & Sippl, 2007*) for comparing determined structure with existing PDB structure and finally Verify 3D (*Lüthy, Bowie & Eisenberg, 1992*) was used to determine compatibility between amino acids and 3D structure.

## Binding site prediction

The topological and geometric properties of the NS1 protein were analyzed using the Computed Atlas of Surface Topography of proteins (CASTp) server (http://sts.bioe.uic.edu/castp/index.html?3trg). This tool identifies, delineates, and measures the pockets and cavities within protein structures. CASTp operates on principles from computational geometry, specifically using alpha shape and pocket algorithms. It employs the molecular surface model (Connolly's surface) and the solvent-accessible surface model (Richards' surface) to quantify the area and volume of each pocket and void in the protein (*Tian et al., 2018*).

## Virtual screening

Virtual screening (VS) helps to find NS1 protein inhibitors from a large library of phytochemicals from the IMPPAT database. The ligands were converted into AutoDock PDBQT format from this SDF file within PyRx 0.8 (*Dallakyan & Olson, 2015*) using Open Babel. The PyRx AutoDock Vina option was used for the virtual screening with an 'exhaustiveness' value of 12 for primary screening and 100 for secondary screening. To define the search space, a grid box was created at X 59.47 Å, Y 57.19 Å, Z 55.68 Å, with a grid point of 30 Å on XYZ dimensions.

## ADMET studies

Drug development requires assessment of drug-likeness and absorption, distribution, metabolism, excretion, and toxicity (ADMET) characteristics. The Swiss ADME server predicted ADME using each molecule's canonical SMILES (*Daina, Michielin &*

*Zoete, 2017*). The ProTox II web server (*Banerjee et al., 2018*) predicted oral toxicity, carcinogenicity, hepatotoxicity, cytotoxicity, and mutagenicity. Drugs are classified into toxicity classes of 1-6 by ProTox II, with class 1 indicating significant danger (LD50 < 5) and class 6 indicating non-toxicity (LD50 > 5000). The top 15 compounds with the best binding energies were tested in ADMET with FDA-approved NS1 protein antivirals.

## Molecular docking

Molecular docking plays a crucial role in the field of computational drug design by evaluating the interactions between the ligand and the active site of the NS1 protein. This includes analyzing the affinity of binding, free energy, and the stability of the ligand-protein complex. Ligands from ADMET screening were docked with NS1 using AutoDock 4.0 implemented in PyRx 0.8 (*Forli et al., 2016*; *Morris et al., 2009*). Receptors were added with hydrogen atoms (polar) and charges (Kollman) within a docking grid of X 59.47 Å, Y 57.49 Å, and Z 55.73 Å, with X 97, Y 97 and Z 72 points per dimension. The Auto Grid 4.0 uses default parameters for the calculation of desolvation, affinity, and electrostatic maps. The Lamarckian genetic algorithm (LGA) examined the arrangements of ligands in 50 runs, with a maximum of 25,000,000 assessments of energy, a population size of 250, and 27,000 generations in each iteration. The parameters like crossover rate of 0.8 and mutation rate of 0.02 were set, utilizing a two-mode crossover. Docking results assessed ligand binding's effect on NS1 affinity, from BIOVIA Discovery Studio 2021 (*Bouback et al., 2021*).

## MD simulation studies

MD simulations using Desmond 2020.1 were conducted on the NS1 protein and its complexes with L2, L3, L5, and dasabuvir in duplicates to check the stability of the protein-ligand complex. Initially, two simulations were conducted in duplicate using identical starting coordinates and velocity seeds to assess consistency under fixed conditions. In response to best practices for statistical independence, a third independent simulation was performed for each protein-ligand complex using the same initial geometry but with initial atomic velocities randomly generated from a Maxwell–Boltzmann distribution at 300 K using a different random seed. Simulations were carried out at 27 °C with the force field (OPLS-2005) and an explicit solvent model with simple point charge (SPC) water molecules. Each protein-ligand system was solvated in a cubic periodic boundary box with at least 10 Å buffer distance between the protein surface and the box edges in all directions. This resulted in a simulation box of approximately 85.64 Å × 85.64 Å × 85.64 Å (actual dimensions depending on the size of the complex). Three $Cl^-$ ions were added for neutralization, resulting in an ionic strength of 2.88 mM at pH 7. Initial equilibration lasted 10 ns under an NVT ensemble to stabilize protein-ligand complexes, followed by an NPT ensemble of 12 ns of equilibration and a minimization run. The ensemble (NPT) employed the Nose–Hoover chain coupling scheme at 1 bar pressure, variable temperature, and a relaxation time of 1.0 ps. Pressure regulation used the Martyna-Tuckerman-Klein chain coupling scheme with a 2 ps relaxation time and a 2 fs time step. The particle mesh Ewald (PME) method was used for long-range electrostatics with a grid spacing of approximately 1 Å per point, and a Coulomb cutoff of 9 Å. The RESPA integrator calculated bonded

**Table 1  BLASTp results of NS1 protein showing homologous sequence.**

| Sr. No | PDB ID | Organism | Max score | e-Value | Identity % | Query cover |
|--------|--------|----------|-----------|---------|-----------|-------------|
| 01 | 5K6K | Zika virus | 298 | $8Xe^{-99}$ | 43.38 | 100% |
| 02 | 4O6C | West Nile virus | 292 | $3Xe^{-96}$ | 40.00 | 100% |
| 03 | 4O6B | Dengue virus 2 | 274 | $2Xe^{-89}$ | 39.44 | 100% |
| 04 | 5YXA | Yellow fever virus | 179 | $5Xe^{-55}$ | 48.88 | 50% |
| 05 | 5O36 | Japanese encephalitis virus | 171 | $4Xe^{-51}$ | 45.36 | 51% |

forces with a 2 fs time step. A 200 ns production run assessed MD simulation stability, monitoring RMSD and RMSF parameters (*Bowers et al., 2006*).

## Binding free energy analysis of protein-ligand complex

The molecular mechanics-generalized born surface area (MM-GBSA) method calculated free energies (binding) for L2, L3, L5, and dasabuvir ligand-protein complexes. A Python script (thermal mmgbsa.py) was used under Prime MM-GBSA to analyze the trajectory of the last 50 frames, considering each frame separately (1 step per frame), resulting in a total of 50 structures, yielding free energies (binding) in kcal/mol. This approach assessed covalent bonds, Coulombic interactions, van der Waals forces, hydrogen bonds, lipophilic interactions, self-contact energy, and protein solvation to evaluate overall ligand-protein interaction stability (*Genheden & Ryde, 2015*).

# RESULTS

## Sequence analysis

The protein sequence of NS1 protein (778–1,130 of 353 aa) with ID D7RF80.1 was retrieved from the UniProt database. BLASTp analysis against the PDB database using default parameters identified homologous sequences, detailed in Table 1. NS1 protein of KFDV is homologous, as indicated by e-value, with Zika virus, West Nile virus, dengue virus-2, yellow fever virus, and Japanese encephalitis virus (PDB ID—5K6K, 4O6C, 4O6B, 5YXA, and 5O36, respectively).

The physicochemical characteristics of the protein NS1 were evaluated through the tool ProtParam, as summarized in Table 2. The theoretical isoelectric point (pI) of NS1 calculated as 7.18, indicates its basic nature, where it achieves electrical neutrality and stability in an electro-focusing system (*Xia, 2007*). Additionally, the aliphatic index (AI) of NS1 of 72.89 suggests stability of NS1 protein across various temperatures due to the presence of aliphatic side chain amino acids (96 amino acids accounting for 27.1%) such as isoleucine (9 No. with 2.5%), alanine (29 No. with 8.2%),, leucine(25 No. with 7.1%), and valine (33 No. with 9.3%) (*Hoda, Tafaj & Sallaku, 2021*).

The NS1 protein's instability index (II) was determined to be 52.92, indicating unstable in solutions containing specific dipeptides (*Guruprasad, Reddy & Pandit, 1990*), as it falls above the 40 threshold. The grand average of hydropathicity (GRAVY) value calculated was −0.291, indicating the protein's hydrophilic nature and favorable affinity towards water
**Table 2  NS1 protein physicochemical characteristics.**

| Features | Remark |
|---|---|
| Protein | NS1 |
| Accession number | D7RF80.1 \| POLG_KFDV:778-1130 |
| Length of sequence | 353 aa |
| Index (Instability) (II) | 52.92 unstable |
| GRAVY | −0.291 |
| Index (Aliphatic) | 72.89 |
| Molecular mass | 38.94 kDa |
| pI (Theoretical) | 7.18 |

**Table 3  Secondary structure analysis of NS1 protein by SOPMA.**

| | SOPMA |
|---|---|
| Alpha helix (Hh) | 74 is 20.96% |
| Extended strand (Ee) | 78 is 22.10% |
| Pi helix (Ii) | 0 is 0.00% |
| Bend region (Ss) | 0 is 0.00% |
| Random coil (Cc) | 185 is 52.41% |
| Beta turn (Tt) | 16 is 4.53% |
| Beta bridge (Bb) | 0 is 0.00% |
| 310 helix (Gg) | 0 is 0.00% |

**Table 4  NS1 protein-Transmembrane region.**

| | Location of domain | Sequence position |
|---|---|---|
| NS1 | extracellular region | 1-353 |
| | TM Helix | 0 |
| | cytoplasmic region | 0 |
| | transmembrane | 0 |

molecules. This is particularly relevant given NS1's role in replication, immune evasion, and pathogenesis (*Muller & Young, 2013*).

The NS1 protein's secondary structure analysis was evaluated using the Structure and Function of Biomolecules: Parallelized In-Silico Molecular Analysis (SOPMA), and the results are detailed in Table 3. The analysis revealed that the NS1 consists of alpha helices, comprising 20.96%, random coils of 52.41%, extended strands of 22.10% and beta turns of 4.53%, of its amino acid composition. This distribution suggests NS1 can be classified as a mixed-class protein (*Lee et al., 2021*).

Furthermore, TMHMM 2.0 predicted the transmembrane region of NS1, classifying it as an extracellular protein, with all amino acids located in this region as detailed in Table 4.

### 3D structure determination and validation

The NS1 protein's 3D structure was determined using I-Tasser-MTD, Robetta, and Swiss Model servers. The model generated by I-Tasser-MTD produced three domain models

(Domain1:1-176 amino acid (AA), Domain 2:177-259AA, and Domain 3:260-353AA), assembled into a single model having a c-score of 2.13, 0.99 ± 0.04 TM-score, and 2.3 ± 1.8Å RMSD was selected. Templates considered for domain prediction were from Zika virus (PDB ID: 5K6K) with 44% identity, dengue virus type 2 (PDB ID: 7WUR) with 39% identity, and Zika virus (PDB ID: 5GS6) with 43% identity. The Swiss Model used the template from Zika virus (PDB ID: 5K6K) with a GMQE score of 0.77 and sequence identity of 43.14%, whereas the Robetta model uses the domain modeling generated the model of confidence score of 0.943. Templates considered for protein prediction were from Zika virus (PDB ID:5K6K and 5GS6 with 41.55% & 43.10% identity, respectively), West Nile virus (PB ID:4O6C and 4TPL with 37.82% & 36.89% identity respectively). To validate these models, a range of assessment methodologies for structure quality were utilized and summarized in Table 5. The stereochemical analysis using PROCHECK revealed different regions within the models, including the most favoured, additional allowed, generously allowed, and disallowed regions. Approximately 88.0% of amino acids in the Swiss model were in the favoured region, indicating higher stereochemical quality compared to I-Tasser MTD and Robetta. Verify3D analysis showed that over 79% of residues in all models had a 3D-1D score of ≥0.2 on average, failing the validation criteria. I-Tasser had a higher ERRAT score (86.09) compared to the Swiss Model (86.07) and Robetta (85.40), indicating better tertiary structure quality. ProSA analysis confirmed Z-scores of the models were within the range of PDB native structures, further validating the quality of the model. In light of this validation, the model generated by I-Tasser, Robetta, and the Swiss Model was selected for energy minimization by the YASARA server, minimizing its energy from −31,525.31 kcal/mol to −42,287.73 kcal/mol for the Swiss Model, −173570.3 kcal/mol to −178,868.9 kcal/mol for the Robetta model and −33,339.72 kcal/mol to −37,741.13 kcal/mol for the I-Tasser Model. After energy minimization, minimized models of the Swiss Model and I-Tasser were subjected for validation, and the overall structure quality, assessed by ERRAT, for I-Tasser, was increased to 94.37, which indicates the better quality of the model and will be used for further assessment. Secondary structure analysis of the refined model was conducted using the Stride web interface and the results are depicted in Table S1 and compared with SOPMA results. SOPMA, a sequence-based predictor, estimated a higher proportion of alpha helices (20.96%) and random coils (52.41%), whereas Spoofing, Tampering, Repudiation, Information Disclosure, Denial of Service (DoS), and Elevation of Privilege (STRIDE), which analyzes atomic coordinates, identified fewer helices (9.06%) and a significantly lower fraction of random coils (22.94%). STRIDE also detected a greater number of beta turns (35.41% *vs.* 4.53%) and additional structural features, such as beta bridges and 310 helices, which were absent in SOPMA predictions. These differences suggest that SOPMA overestimates flexible regions, while STRIDE provides a more accurate representation of the protein's structural organization in its folded state. Figures 2 and 3 illustrates the diagrams of the 3D structure and its validation of the selected minimized I-Tasser model. In Fig. 2A, the I-Tasser model was represented in red colour, energy energy-minimized model was represented in blue colour, and the 3D structure of the homologous protein of Zika virus (PDB ID: 5K6K) in yellow colour was superimposed using Discovery Studio software. The overlay similarities between both the I-Tasser model
**Table 5  NS1 Generated model's quality assessment.**

| Validation method | I-Tasser model | Swiss model | Robetta model | Refined I-Tasser model | Refined Swiss model | Refined Robetta model |
|---|---|---|---|---|---|---|
| ERRAT score | 86.095 | 86.07 | 85.40 | 94.37 | 90.90 | 88.72 |
| Procheck | | | | | | |
| Most favoured region | 71.3% | 88.00% | 87.0% | 83.0% | 88.3% | 88.3% |
| Additionally allowed regions | 24.3% | 10.7% | 12.7% | 15.3% | 10.7% | 11.0% |
| Generously allowed region | 3.00% | 0.7% | 0.0% | 0.7% | 0.3% | 0.0% |
| Disallowed region | 1.3% | 0.7% | 0.3% | 1.0% | 0.7% | 0.7% |
| ProSA web score | −6.66 | −7.13 | −7.48 | −7.1 | −6.82 | −7.26 |
| Verify 3D | 77.05% | 78.75% | 76.49% | 76.77% | 75.07% | 76.20% |
| QMean disco global | 0.67 ± 0.05 | 0.70 ± 0.05 | 0.73 ± 0.05 | 0.69 ± 0.05 | 0.70 ± 0.05 | 0.70 ± 0.05 |
| QmeanZscore | −7.96 | −2.49 | 0.14 | −2.95 | −1.74 | −1.34 |
| Qmean all atom | −1.71 | −1.97 | −1.97 | −2.05 | −1.64 | −1.85 |
| Qmean torsion | −6.35 | −1.29 | 1.52 | −1.67 | −0.62 | −0.11 |
| Qmean solvation | −4.01 | −2.06 | −2.56 | −2.40 | −2.03 | −2.45 |
| Qmean Cβ | −1.65 | −2.29 | −1.99 | −2.16 | −1.90 | −1.62 |

and the minimized model were 0.81236, and the RMSD value for C-alpha was 0.54479. Similarly, the overlay similarities between the minimized model and Zika virus NS1 protein were 0.5658, which indicates the 3D structure is highly similar. Figure 3A represents the per-residue error values for a specific protein, covering residues 5 to 349, highlighting regions with potential structural deviations. In contrast, Fig. 3B analyze validation metrics on a broader scale. F 2C presents a Z-score distribution across different protein sizes (up to 1,000 residues), benchmarking the model against experimentally solved structures. A dark dot is placed in Fig. 2C at $x = 353$, indicating the Z-score of the selected protein, aligning it with other proteins of similar size. Additionally, Fig. 3C displays residue-wise validation scores, and are consistent with the overall model assessment.

## Active site determination
The binding pockets (top five) of the NS1 protein were determined by the CASTp server, as represented in Fig. 4 and detailed in Table 6. The pocket panels highlight several residues within these pockets, which could be potential drug targets. Figure 4 illustrate different binding pocket, showing various amino acid residues in the NS1 protein. Table 6 provides CASTp statistics, including the surface-accessible (SA) area and surface-accessible (AS) volume for each pocket. Notably, the top binding pocket of the NS1 protein exhibits the highest surface area of 750.14 Å² and a volume of 725.19 Å³ with a grid box dimension of X 59.47, Y 57.19, and Z 55.68 determined by discovery studio were considered and were further used for the virtual screening of phytochemicals from databases.

## Virtual screening
Virtual screening of desired phytochemicals as inhibitor from chemical database determines the success of *in silico* drug designing approach (*Talevi, 2023*). Hence, three phase virtual screening of phytochemicals was carried from IMPPAT 2.0 database against NS1 protein.

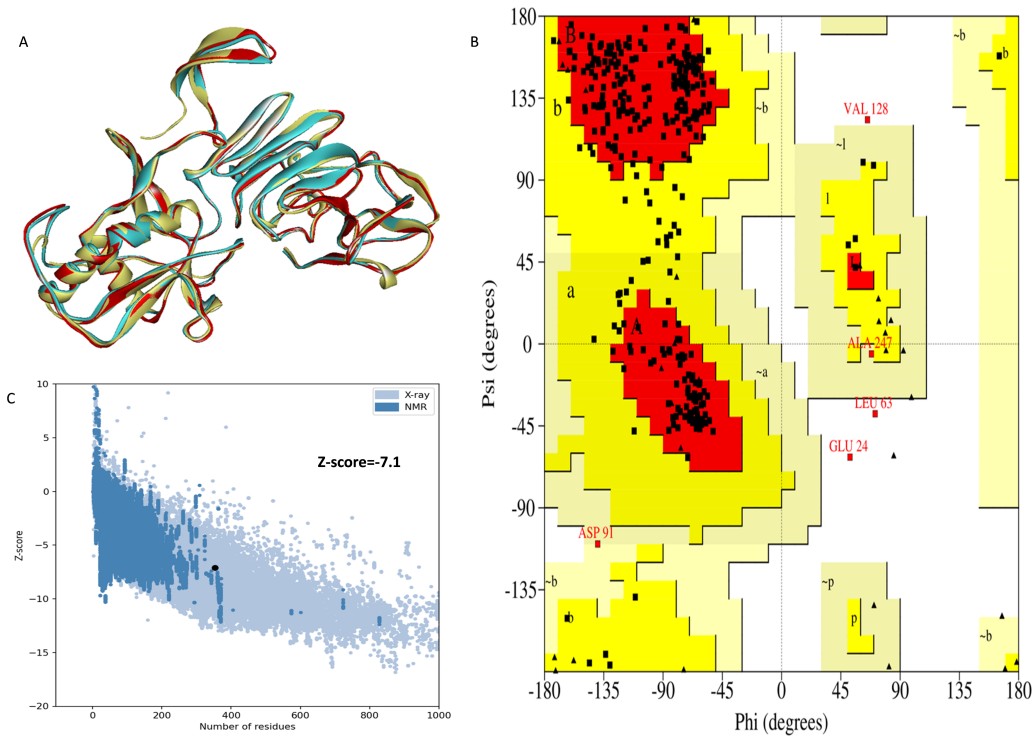

**Figure 2** (A) Super imposed 3D structure of I-Tasser model (red), and minimized model (blue), and Zika virus NS1 protein (yellow) and Validation-NS1 protein; (B) Ramachandran plot; (C) ProSA Web score.

**Table 6** Information of NS1 protein's active site by CastP.

| Predicted binding site | Area (Å2) | Volume (Å3) | Amino acid residue |
|---|---|---|---|
| 1 | 750.14 | 725.19 | Asp1, Met2, Gly3, Cys4, Ala5, Ile6, Asp7, Ala8, Asn9, Arg10, Arg14, Gly16, Glu17, Gly18, Leu19, Val20, Val21, Trp22, Arg23, Glu24, Asp27, Tyr32, Phe160, Gly161, Met162, Thr166, Gly183, Val184, Met185, Gly186, Ala188, Val189, Lys190, Ser191, His193, Ala194, Val195, Thr197, Asp198, Gln199, Trp202 |
| 2 | 273.22 | 125.30 | Phe34, Gln60, Asn61, Arg62, Leu63, Glu64, Arg99, Gly100, Gly101, Lys102, Gly104, Thr105, Thr152, Gly153, Val154, Phe155, Thr156, Val157, Leu170, Arg173, Thr175, Ala176, Ser177, Ser178, Asp179, Cys180, Asp181, Val184, Gln199, Arg223, Cys225 |
| 3 | 146.83 | 41.44 | Pro39, Ser40, Ala73, Val74, Asn76, Leu77, Ala78, Glu81, Met113, Thr115, Ser116, Trp117, Gly119, Gln122, Phe124, Val125, Trp126, Ser127, Val128 |
| 4 | 49.85 | 37.85 | Val25, Thr26, Asp27, Trp28, Ile218, Thr220, Arg276 |
| 5 | 81.95 | 32.77 | Ile57, Val58, Pro59, Lys92, Arg93, Asp94, Gly137, Val139, Glu140, Ala142, Cys145, Leu147, Arg150 |

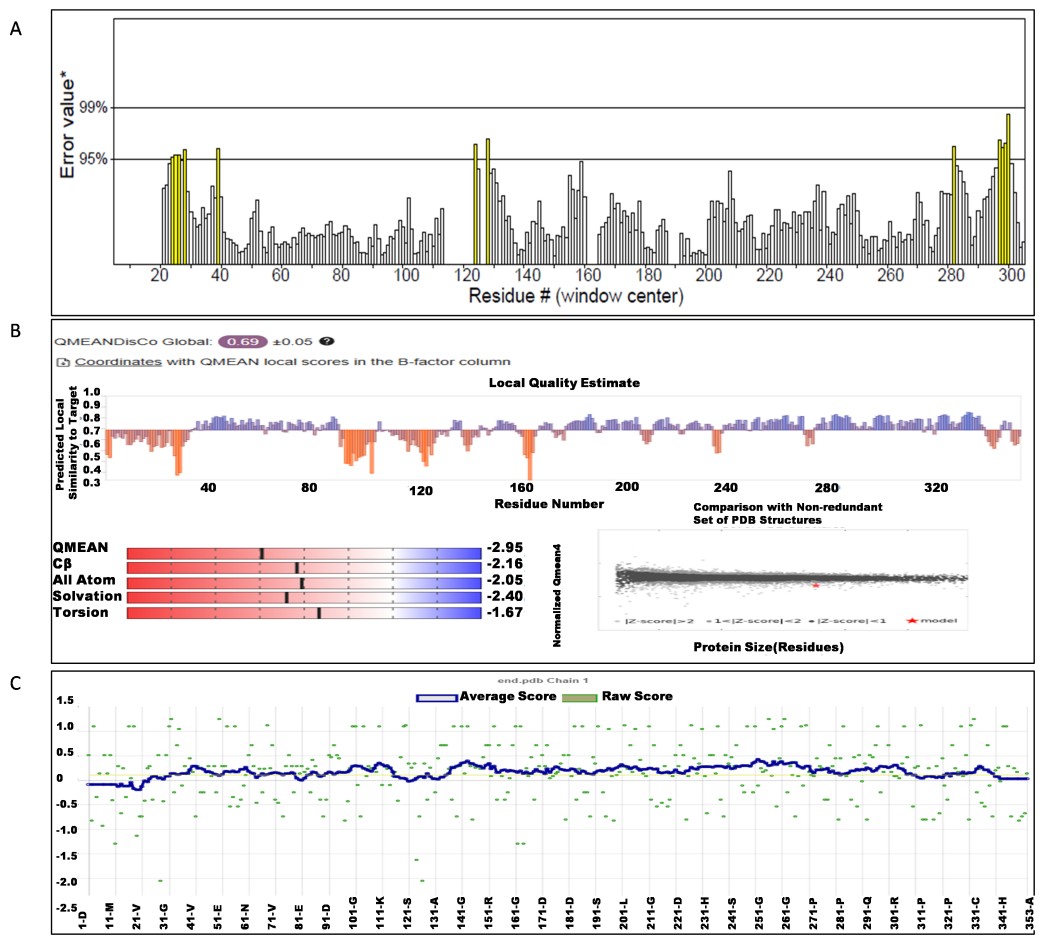

**Figure 3** (A) ERRAT quality factor; (B) Qmean score; and (C) Verify3D score.

The Lipinski RO5 filter was applied for initial selection of compounds of 11,530 from a total of 18,000 phytochemicals. Virtual screening against 11,530 phytochemicals was carried against NS1 protein using PyRx 0.8 with AutoDock Vina configured to an 'exhaustiveness' value of 12 in the first phase using the grid box dimensions X 59.47, Y 57.19, Z 55.68. Based on the binding affinity, the top 10% of the compounds, approximately 1150, were considered for secondary screening using AutoDock Vina with an 'exhaustiveness' value of 100. Based on the binding affinity, further top 10% of compounds, approximately 115, were considered for tertiary screening using AutoDock software. Tables S1 and S3 in the supplementary show a list of selected compounds after secondary and tertiary screening with binding affinity values. A similar approach was carried out by *Joshi et al. (2023)* for the identification of a novel phytochemical inhibitor of DLL3. *Gurung et al. (2020)* predicted potential inhibitors for corona virus from virtual screening of phytochemicals from IMPPAT database. *Saha et al. (2024)*; *Aljahdali, Molla & Ahammad (2021)*; *Roshni et al. (2022)* and *Kalaria & Patel (2020)* identified potential inhibitors from IMPPAT database for Nipah virus, walleye dermal sarcoma virus, SARS-COV-2 and COVID-19 respectively.

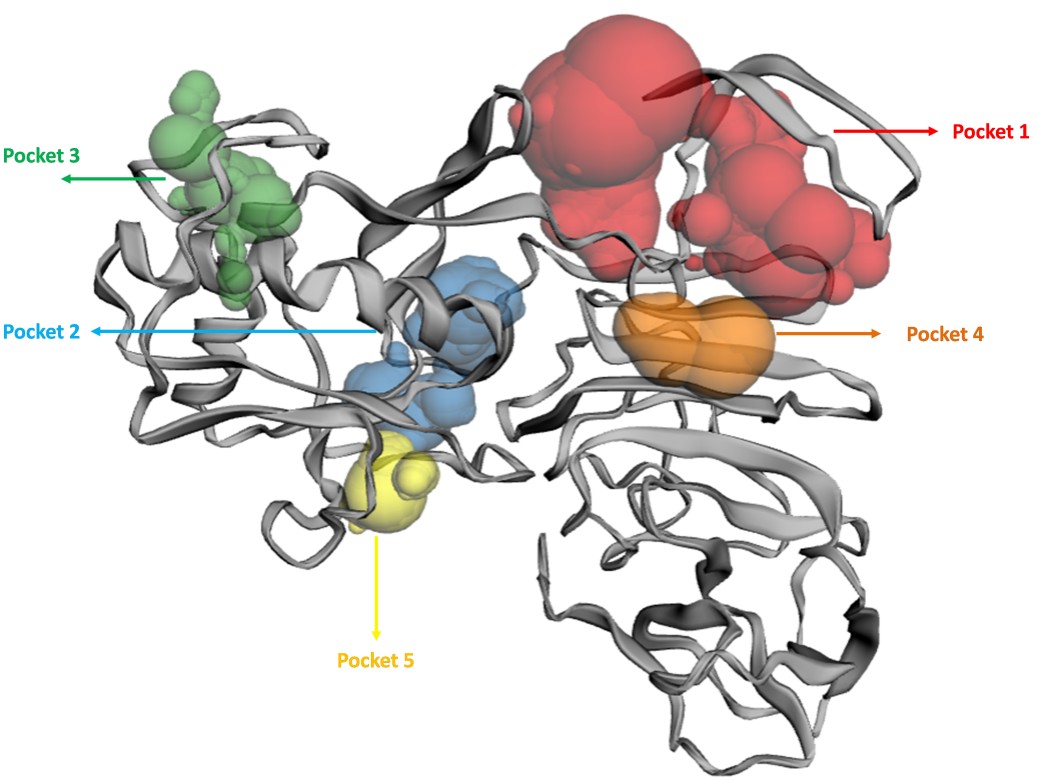

**Figure 4** Active site identified by CastP server.

## Molecular docking

Molecular docking is an essential tool for understanding the precise interaction at the atomic level between ligands and proteins (*Sahu et al., 2024*). It has a crucial function in the evaluation of pharmaceuticals from databases. Following the virtual screening, 115 compounds underwent a docking process using AutoDock 4.2 implemented in PyRx 0.8 was utilized to analyze the binding affinity. The top 15 ligands with the highest binding affinities, as indicated in Table 7, were considered for ADMET.

## Toxicity measurement

*In silico* toxicity studies is essential in drug development before the drug proceeds to clinical trials. The use of *in silico* toxicity evaluations has become more common because of their high level of accuracy, fast processing speed, and easy accessibility. These evaluations allow for a thorough examination of both synthetic and natural substances. We employed the ProTox II server to evaluate the toxicity and possible negative impacts of fifteen specific compounds and compared them with standard FDA-approved antivirals. The evaluation encompassed a range of toxicological factors, including carcinogenicity, hepatotoxicity, acute toxicity, cytotoxicity, and mutagenicity, and the calculation of the LD50 (median lethal dosage) in milligrams per kilogram. These findings are displayed in Table 8. Based on the findings by ProTox II, compounds classified in class 4 or higher were further evaluated for specific types of toxicity. Compounds except IMPHY000366 (L1),

Achappa et al. (2025), *PeerJ*, DOI 10.7717/peerj.19954

**Table 7 Results of molecular docking by AutoDock.**

| Compound ID | Structure | Name | Name of plant | Binding affinity Autodock (kcal/mol) | Binding affinity Autodock vina (kcal/mol) | Inhibition contant Ki (μM) |
|---|---|---|---|---|---|---|
| IMPHY000366 | | Jervine | *Veratrumviride* | −9.76 | −9.00 | 0.070 |
| IMPHY010294 | | Solanogantamine | *Solanum giganteum* | −9.34 | −9.80 | 0.131 |
| IMPHY001281 | | Ranmogenin A | *Rohdeanepalensis* | −9.12 | −8.80 | 0.083 |
| IMPHY004082 | | Peiminine | *Fritillariaimperialis* | −9.08 | −8.80 | 0.222 |
| IMPHY011162 | | Solagenin | *Solanum asperolanatum* | −9.08 | −8.80 | 0.222 |
| IMPHY003352 | | Murrayazolinol | *Murrayakoenigii* | −8.99 | −9.00 | 0.256 |
| IMPHY001309 | | Anabsinthin | *Artemisia absinthium* | −8.94 | −8.60 | 0.281 |
| IMPHY010476 | | Withametelin F | *Datura metel* | −8.93 | −9.40 | 0.286 |
| IMPHY001575 | | 9-Dehydrohecogenin | *Agave cantala* | −8.92 | −9.20 | 0.289 |
| IMPHY008834 | | Withaphysalin C | *Physalisindica* | −8.92 | −9.00 | 0.290 |

**Table 7** (*continued*)

| Compound ID | Structure | Name | Name of plant | Binding affinity Autodock (kcal/mol) | Binding affinity Autodock vina (kcal/mol) | Inhibition contant Ki (μM) |
|---|---|---|---|---|---|---|
| IMPHY001343 | | Nodosin | *Cassia javanica* | −8.85 | −8.80 | 0.327 |
| IMPHY009308 | | Pennogenin | *Paris polyphylla* | −8.83 | −9.20 | 0.334 |
| IMPHY008030 | | Hispidogenin | *Solanum asperolanatum* | −8.83 | −8.80 | 0.336 |
| IMPHY004035 | | Ruscogenin | *Cordylineaustralis* | −8.70 | −8.80 | 0.416 |
| IMPHY004234 | | Neochlorogenin | *Solanum asperolanatum* | −8.70 | −8.80 | 0.419 |
| 56640146 | | Dasabuvir FDA approved antiviral | | −8.00 | | 1.37 |

**Table 8  ProTox II: Selected drugs toxicity determination.**

| Name of drug | Oral toxicity | | Organ toxicity-Hepatotoxicity | Carcinogenicity | Mutagenicity | Cytotoxicity |
|---|---|---|---|---|---|---|
| | LD50 | Class | | | | |
| IMPHY000366 (L1) | 260 | III | ✗ | ✗ | ✗ | ✗ |
| IMPHY010294 (L2) | 3500 | V | ✗ | ✗ | ✗ | ✗ |
| IMPHY001281 (L3) | 8000 | VI | ✗ | ✗ | ✗ | ✗ |
| IMPHY004082 (L4) | 280 | III | ✗ | ✗ | ✗ | ✗ |
| IMPHY011162 (L5) | 10000 | VI | ✗ | ✗ | ✗ | ✗ |
| IMPHY003352 (L6) | 750 | IV | ✗ | ✗ | ✗ | ✗ |
| IMPHY001309 (L7) | 452 | IV | ✗ | ✗ | ✗ | ✗ |
| IMPHY010476 (L8) | 300 | III | ✗ | ✔ | ✗ | ✗ |
| IMPHY001575 (L9) | 400 | IV | ✗ | ✗ | ✗ | ✗ |
| IMPHY008834 (L10) | 100 | III | ✗ | ✗ | ✗ | ✔ |
| IMPHY001343 (L11) | 123 | III | ✗ | ✗ | ✗ | ✔ |
| IMPHY009308 (L12) | 8000 | VI | ✗ | ✗ | ✗ | ✗ |
| IMPHY008030 (L13) | 2600 | V | ✗ | ✗ | ✗ | ✗ |
| IMPHY004035 (L14) | 8000 | VI | ✗ | ✗ | ✗ | ✗ |
| IMPHY004234 (L15) | 2600 | V | ✗ | ✗ | ✗ | ✗ |
| **FDA approved antivirals** | | | | | | |
| Dasabuvir | 1000 | IV | ✔ | ✗ | ✗ | ✗ |
| Favipiravir | 1717 | IV | ✗ | ✔ | ✗ | ✗ |
| Galidesivir | 500 | IV | ✗ | ✗ | ✗ | ✗ |
| Remdesivir | 1000 | IV | ✗ | ✗ | ✗ | ✗ |
| Ribavirin | 2700 | V | ✗ | ✔ | ✗ | ✗ |
| Sofosbuvir | 12000 | VI | ✔ | ✗ | ✗ | ✗ |

**Notes.**
✗, Inactive.
✔, Active.

IMPHY004082 (L4), IMPHY010476 (L8), IMPHY008834 (L10), and IMPHY009308 (L11) are classified in class 4 and above, showed no oral toxicity, carcinogenicity, mutagenicity, or cytotoxicity. Subsequently, based on the ProTox II results and AutoDock results, compounds IMPHY010294 (L2), IMPHY001281 (L3), IMPHY011162 (L5), IMPHY003352 (L6), IMPHY001309 (L7), and IMPHY001575 (L9) were chosen for further analysis focusing on absorption, distribution, metabolism, and excretion (ADME) properties.

## ADME analysis

In order to have therapeutic benefits, a drug candidate must reach the target location in the body at the most effective concentrations (*Subbaiah, Rautio & Meanwell, 2024*). Anticipating pharmacological and physicochemical factors is essential as it offers valuable information about whether a molecule can effectively reach and sustain enough concentrations at the desired site to initiate the intended biological reaction. The six compounds with FDA approved dasabuvir underwent evaluation of their physicochemical and pharmacokinetics attributes using SwissADME (*Tataringa et al., 2024*) and are summarized in Table 9. This evaluation involved assessing compliance with Veber's

Rule and Lipinski's Rule of Five (RO5) to ascertain their suitability for oral activity (*Lorin et al., 2024*). Table 9 provides a summary of all the chosen compounds passed both Lipinski and Veber's rule requirements. Lipophilicity, which enhances the solubility of drug-like substances in fats and oils, increases their dispersion through cell membranes, indicating the possibility for oral delivery. All drugs demonstrated high gastrointestinal absorption compared to FDA approved dasabuvir drug, indicating that drugs can be given as tablet formulation. All six compounds exhibited desirable lipophilicity, as shown by a Consensus Log $P$ value of around 3.5, which falls within the range of +5.00. All the drugs exhibited moderate solubility when compared to dasabuvir (poorly soluble), which is an important factor that affects their bioavailability and bioactivity (*Kibet et al., 2024*). The assessment of molar refractivity (MR) gives valuable information on the pharmacodynamics and pharmacokinetics of the compounds, considering various interactions in solution such as drug-co-solute, drug-drug, and drug-solvent interactions (*Sakyi et al., 2022*). It was observed that the selected drugs fall into the ideal MR range of 40 to 130. The evaluations of topological polar surface area (TPSA) suggested that all compounds having TPSA <90 Å$^2$ indicating high oral bioavailability, absorption, and high transport properties. The synthetic accessibility (*Ertl & Schuffenhauer, 2009*) score typically range from 1 (easy synthesis) to 10 (very difficult to synthesize). The compound having score less than 6 is considered as good synthetic accessible drug. All the drugs except L2 and L5 have low synthetic accessible score. The search for possible non-specific interaction with biological targets led to the identification of PAINS (Pan Assay INterference chemicals) (*Xu et al., 2024*; *Nasiri et al., 2017*). All the compounds were identified of not having PAINS substructures and it falls inline with the FDA approved dasabuvir drug. Pharmacokinetic studies evaluated criteria such as the capacity of a substance to pass through the blood–brain barrier (BBB), its absorption in the gastrointestinal tract (GI), and its interactions with permeability glycoprotein (P-gp) (*Yu, 1999*). All the compounds except L3 and L6 would be able to pass across the blood–brain barrier (*Lee et al., 2024*). All compounds demonstrated high gastro intestinal absorption rate compared to FDA approved dasabuvir drug (*Schanker, 2022*). Additionally, it was projected that all compounds would serve as substrates for P-glycoprotein, which might potentially increase their bioavailability. Overall, our forecasts suggest that all substances are likely to have advantageous pharmacokinetic and pharmacodynamic characteristics.

## NS1-ligand interaction analysis

Determining the amino acids of protein involved in the interaction with drug is essential to determine its biological activity (*Khan et al., 2024*). The development of pharmaceuticals and the management of disease both heavily depend on these interactions. Consequently, the interaction between the ligands and the NS1 protein was comprehensively analyzed using BIOVIA Discovery Studio Visualizer. The findings of this investigation of interactions, which encompass HB (carbon HB, conventional, and pi-donor H-B), diverse HP contacts (pi-pi T-shaped, alkyl, pi-alkyl, and pi-sigma), and electrostatic interactions (pi-anion), are displayed in Table 10 and Figs. 5 and 6.

**Table 9  Swiss ADME: selected drugs ADME characterization.**

| Molecule (IMPHY) | 010294 (L2) | 001281 (L3) | 011162 (L5) | 003352 (L6) | 001309 (L7) | 001575 (L9) | Dasabuvir |
|---|---|---|---|---|---|---|---|
| Formula | C27H46N2O | C27H42O6 | C27H42O4 | C23H25NO2 | C30H40O6 | C27H40O4 | C26H27N3O5S |
| Physico-chemical properties | | | | | | | |
| Weight (molecular) (Da) | 414.67 | 462.62 | 430.62 | 347.45 | 496.64 | 428.6 | 493.6 |
| #Heavy atoms | 30 | 33 | 31 | 26 | 36 | 31 | 35 |
| #Rotatable bonds | 0 | 0 | 0 | 0 | 0 | 0 | 6 |
| #Heavy atoms (Aromatic) | 0 | 0 | 0 | 13 | 0 | 0 | 22 |
| #H-bond donors | 2 | 4 | 1 | 1 | 1 | 1 | 2 |
| #H-bond acceptors | 3 | 6 | 4 | 2 | 6 | 4 | 5 |
| MR | 129.38 | 125.12 | 122.27 | 105.52 | 133.61 | 121.79 | 139.63 |
| TPSA | 49.49 | 99.39 | 55.76 | 34.39 | 82.06 | 55.76 | 118.64 |
| Consensus Log P (Lipophilicity) | 3.41 | 3.38 | 3.92 | 3.34 | 3.76 | 4.09 | 3.80 |
| Water solubility ESOL Log S | −5.99 | −4.28 | −5.49 | −5.00 | −4.59 | −5.31 | −5.65 |
| Water solubility ESOL Class | Moderate | Moderate | Moderate | Moderate | Moderate | Moderate | Poor |
| Pharmacokinetics | | | | | | | |
| GI absorption | High | High | High | High | High | High | Low |
| log Kp (cm/s) skin permeation | −4.8 | −7.35 | −5.57 | −5.45 | −7.45 | −5.75 | −6.29 |
| BBB permeate | ✔ | X | ✔ | ✔ | X | ✔ | X |
| Drug likeness | | | | | | | |
| Lipinski #violations | 0 | 0 | 0 | 0 | 0 | 0 | 0 |
| Veber #violations | 0 | 0 | 0 | 0 | 0 | 0 | 0 |
| Bioavailability Score | 0.55 | 0.55 | 0.55 | 0.55 | 0.55 | 0.55 | 0.56 |
| PAINS #alerts | 0 | 0 | 0 | 0 | 0 | 0 | 0 |
| Leadlikeness #violations | 2 | 1 | 2 | 1 | 1 | 2 | 2 |
| Synthetic Accessibility | 5.3 | 7.02 | 6.68 | 4.82 | 7.08 | 6.77 | 3.46 |
| Metabolism | | | | | | | |
| Pgp substrate | ✔ | ✔ | ✔ | ✔ | ✔ | ✔ | X |
| Inhibitor | | | | | | | |
| CYP2D6 | X | X | X | ✔ | X | X | X |
| CYP2C9 | X | X | X | X | X | X | ✔ |
| CYP2C19 | X | X | X | X | X | X | ✔ |
| CYP1A2 | X | X | X | X | X | X | ✔ |
| CYP3A4 | X | X | X | X | X | X | X |

**Notes.**
X, No.
✔, Yes.

The study of interaction between NS1- L2 complex (Figs. 5A and 5B), NS1- L3 complex (Figs. 5C and 5D), NS1- L5 complex (Figs. 5E and 5F), NS1- L6 complex (Figs. 6A and 6B), and NS1-dasabuvir complex (Figs. 6C and 6D) was studied. The complex NS1-L2 has a total of eight bonding interactions were identified with a binding energy of −9.34 kcal/mol. These included five HB, consisting of three conventional bonds (CHB) and two carbon HB bond, as well as three hydrophobic (HP) contact known as an alkyl interaction. The amino acid residues Glu159, Glu24, Trp22, and Met185 formed typical

HB with the ligand. Additionally, Val21, Leu19, and Phe160 participated in hydrophobic interaction especially an alkyl association with the ligand.

For NS1-L3 complex a total of 11 bonding interactions with a binding energy of −9.12 kcal/mol. These interactions include six hydrogen bonds (five CHB and one carbon HB bond) as well as 5 HP interactions (three alkyl and two pi-alkyl). Amino acids Gln199, His231, Glu24, and Gly183 established CHB with the ligand through H−O interactions, whereas Ala187 participated in carbon-HB. Val21 formed two alkyl connections and Leu19 formed one alky interaction, whereas His231 participated in two pi-alkyl interactions with the ligand.

For NS1-L5 complex, a total of four bonding interactions were identified with a binding energy of −9.08 kcal/mol. These interactions include one HB (conventional), and three HP contacts (two alkyl and one pi-alkyl). The residues Met185 engaged in CHB with the ligand through H−O interactions, whereas Met162 established two alkyl interactions with the ligand and Phe160 involved in pi-alkyl interactions with ligand.

Similarly for NS1- L6 complex a total of 10 bonding interactions were formed with a binding energy of −8.99 kcal/mol. These interactions include one HB (CHB), and nine HP interactions (two amide, six alkyl, and one pi-alkyl). The ligand produced CHB with the amino acids Glu24. Additionally Val184 and Met185 formed amide interactions with ligand, whereas Val21 (three alkyl interaction), Leu19 (two alkyl interaction), and Val184 (one alkyl interaction) established hydrophobic interaction with ligand. Whereas amino acid Tyr32 formed pi-alkyl interaction with ligand.

To compare the interaction analysis of the ligand from IMPPAT database with the FDA approved antiviral, the interaction analysis between NS1-dasabuvir was studied. Although Dasabuvir specifically targets the NS5B polymerase and not NS1, it was included in the study as a reference compound to enable comparative evaluation against an FDA-approved antiviral agent with activity against flaviviral non-structural proteins. This interaction demonstrated a total of 12 bonding interactions. The molecule forms a total of four HB, consisting of three CHB, and 1 carbon HB. Additionally, there are one additional interactions involving electrostatic bonding. Furthermore, there are seven HP contacts, including two pi-pi T-shaped interaction, one pi-sigma, one pi-pi-stacked interaction, one pi-pi-T-shaped interaction and three pi-alkyl interactions. The interactions exhibited a binding energy of −8.0 kcal/mol. significantly, the amino acids Gly163, Gln199, and Glu24 established CHB with the ligand, whereas residues Met162 involved in H−O carbon-HB, and Glu24 produced an electrostatic pi-anion interaction with ligand. The ligand formed pi-sigma interactions with Thr166, pi-pi T-shaped interactions with Tyr32 and Phe160, pi-pi-stacked interaction with Phe160, and pi-alkyl interactions with Phe160 and Val184. After analyzing this interaction, we selected ligands L2, L3, and L5 together with dasabuvir for further simulation studies.

## Molecular dynamics simulation studies
### NS1 protein
To assess the stability and convergence of the NS1 protein, molecular dynamics (MD) simulations were performed over 200 ns in duplicates. The root mean square deviation

Achappa et al. (2025), *PeerJ*, DOI 10.7717/peerj.19954

**Table 10  NS5 protein-ligand interaction analysis.**

| Complex | Amino acid residue | Bond distance (Å) | Bond category | Type of bond | Chemistry |
|---|---|---|---|---|---|
| L2 | Glu159 | 2.22 | HB | CHB | Donor |
| | Glu159 | 1.92 | HB | CHB | Donor |
| | Glu24 | 2.03 | HB | CHB | Acceptor |
| | Trp22 | 3.48 | HB | CHB | Acceptor |
| | Met185 | 3.69 | HB | CHB | Acceptor |
| | Val21 | 5.12 | HB | Carbon HB | Acceptor |
| | Leu19 | 4.19 | HP | Alkyl | Alkyl |
| | Phe160 | 5.25 | HP | Alkyl | Alkyl |
| L3 | Gln199 | 2.50 | HB | CHB | Donor |
| | His231 | 2.88 | HB | CHB | Donor |
| | Glu24 | 1.80 | HB | CHB | Acceptor |
| | Gly183 | 1.76 | HB | CHB | Acceptor |
| | Glu24 | 1.70 | HB | CHB | Acceptor |
| | Ala187 | 3.64 | HB | Carbon-HB | Acceptor |
| | Val21 | 5.07 | HP | Alkyl | Alkyl |
| | Val21 | 5.49 | HP | Alkyl | Alkyl |
| | Leu19 | 5.04 | HP | Alkyl | Alkyl |
| | His231 | 5.21 | HP | Pi-Alkyl | Pi-Orbitals |
| | His231 | 5.02 | HP | Pi-Alkyl | Pi-Orbitals |
| L5 | Met185 | 2.01 | HB | CHB | Acceptor |
| | Met162 | 4.67 | HP | Alkyl | Alkyl |
| | Met162 | 4.39 | HP | Alkyl | Alkyl |
| | Phe160 | 4.35 | HP | Pi-Alkyl | Pi-Orbitals |
| L6 | GLU24 | 1.82 | HB | CHB | Acceptor |
| | VAL184 | 3.81 | HP | Amide Pi-Stacked | Amide |
| | MET185 | 3.81 | HP | Amide Pi-Stacked | Amide |
| | VAL21 | 5.09 | HP | Alkyl | Alkyl |
| | LEU19 | 4.93 | HP | Alkyl | Alkyl |
| | VAL21 | 4.56 | HP | Alkyl | Alkyl |
| | LEU19 | 3.80 | HP | Alkyl | Alkyl |
| | VAL21 | 4.79 | HP | Alkyl | Alkyl |
| | VAL184 | 4.32 | HP | Alkyl | Alkyl |
| | TYR32 | 4.34 | HP | Pi-Alkyl | Pi-Orbitals |

*(continued on next page)*

Achappa et al. (2025), *PeerJ*, DOI 10.7717/peerj.19954

**Table 10** (*continued*)

| Complex | Amino acid residue | Bond distance (Å) | Bond category | Type of bond | Chemistry |
|---------|--------------------|-------------------|---------------|--------------|-----------|
| | GLY163 | 2.08 | HB | CHB | Donor |
| | GLN199 | 2.79 | HB | CHB | Donor |
| | GLU24 | 2.12 | HB | CHB | Acceptor |
| | MET162 | 3.03 | HB | Carbon HB | Donor |
| | GLU24 | 3.96 | Electrostatics | Pi-Anion | Negative |
| Dasabuvir | THR166 | 3.75 | HP | Pi-Sigma | C-H |
| | PHE160 | 4.32 | HP | Pi- Pi-Stacked | Pi-Orbitals |
| | TYR32 | 5.76 | HP | Pi-Pi T-Shaped | Pi-Orbitals |
| | PHE160 | 5.00 | HP | Pi-Pi T-Shaped | Pi-Orbitals |
| | PHE160 | 5.05 | HP | Pi-Alkyl | Pi-Orbitals |
| | PHE160 | 5.02 | HP | Pi-Alkyl | Pi-Orbitals |
| | VAL184 | 5.43 | HP | Pi-Alkyl | Pi-Orbitals |

**Notes.**

HB, Hydrogen Bond; HP, Hydrophobic; CHB, Conventional H-B.

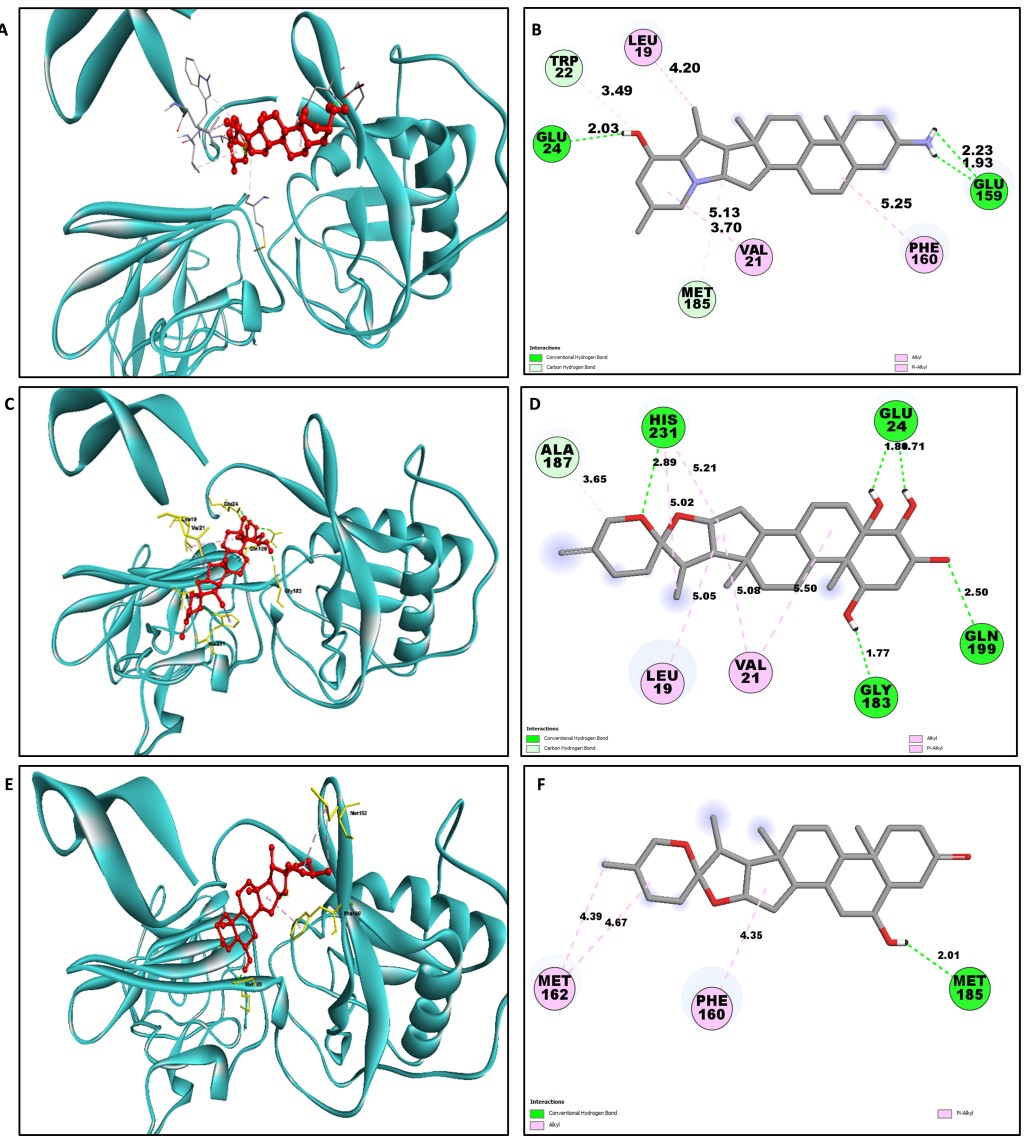

**Figure 5** NS1 protein (blue ribbon model)-ligand (ball and stick (red)) 3D interaction analysis (A, C, E) and 2D interaction plot (stick model) (B, D, F).

(RMSD) of the NS1 protein started at 1.76 Å at 0 ns, gradually increased in a step from 2.8 Å at 5 ns, 3 Å at 10 ns, 3.5 Å at 25 ns, 3.8 Å at 30ns, and 4.2 Å at 50 ns. Further, RMSD remained stable at 4 Å from 50 ns to 160 ns. After 160 ns RMSD increased to 4.648 Å and remained stable up to 200 ns (Fig. 7A). This RMSD range is acceptable in the range 1.76 to 4.648 Å in duplicates, indicating no major conformational changes. The average RMSD of NS1 protein was found to be 3.71 ± 0.44 Å. Stable RMSD values suggest good simulation convergence and stable protein conformations. The root mean square fluctuations (RMSF) plot showed notable fluctuations in regions other than the $\alpha$-helix and $\beta$-sheets, with more fluctuation at the N and C-terminals and also between Gly100-Arg150 amino acid (Fig. 7B).

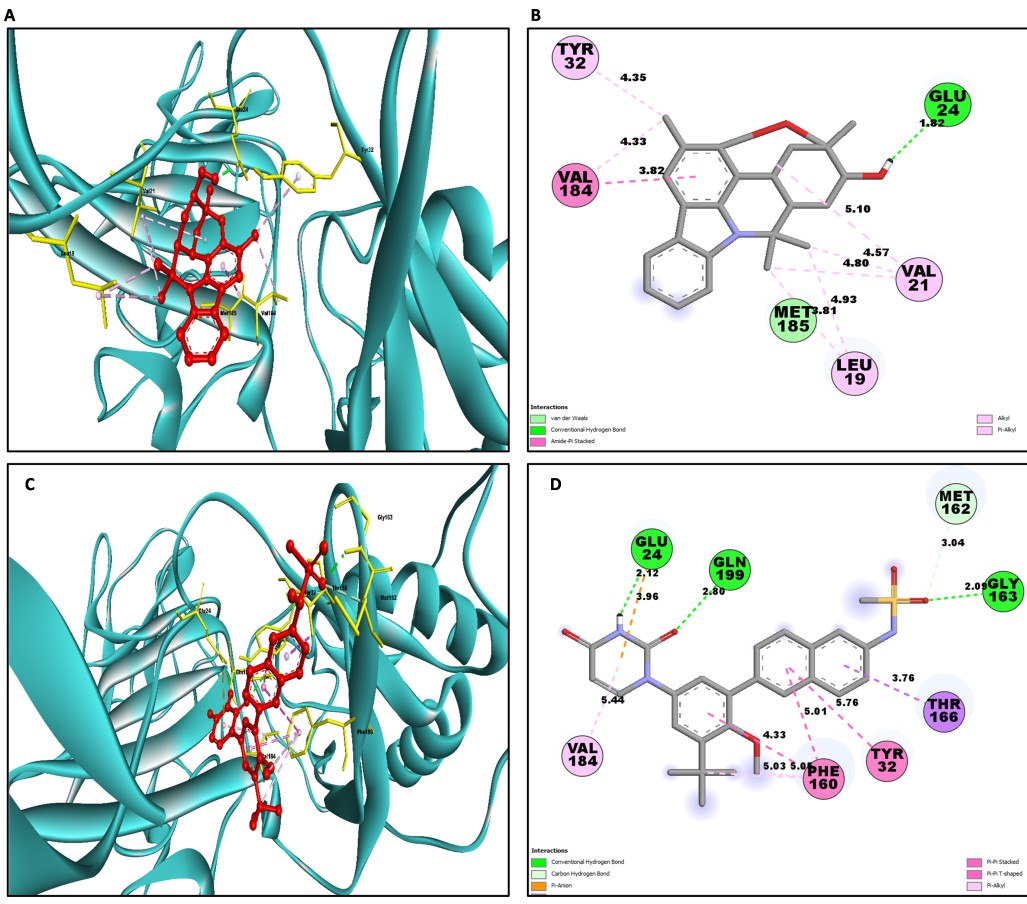

**Figure 6** NS1 protein (blue ribbon model)-ligand (ball and stick (red)) 3D interaction analysis (G, I) and 2D interaction plot (stick model) (H, J).

The RMSF figures indicate that the protein structure maintains its rigidity throughout the simulation but due to less secondary structural element (30.01%), more fluctuations were observed in the regions of coils and turns. The RMSF was fluctuated between 0.576 to 5.44 Å both in duplicates and an average RMSF value found to be 1.60 ± 0.910 Å. The RMSF graphic includes annotations for secondary structural features, including alpha-helical areas (highlighted in red) and beta-stranded regions (highlighted in blue).

## RMSD of NS1 protein-Ligand complex

The NS1 protein-ligand complex's RMSD trajectory is shown graphically in Fig. 8. The accompanying image shows the protein's RMSD trajectory in blue, along with the associated RMSD values displayed in Å units on the left side of the $Y$-axis. Furthermore, the ligand's RMSD trajectory is shown in red in the figure, along with the matching values in Å units on the $Y$-axis (right). The RMSD of the NS1-L2 complex (Fig. 8A) started at 0 ns at approximately 3.0 Å and rising to 3.5 Å at 10 ns and maintained up to 25 ns. After that, the protein's RMSD trajectory raised, reaching 4.0 Å at 30 ns and remained stable by fluctuating between 4.0 Å to 4.2 Å from 30 ns to 175 ns. At the end of 175 ns MD simulation, the
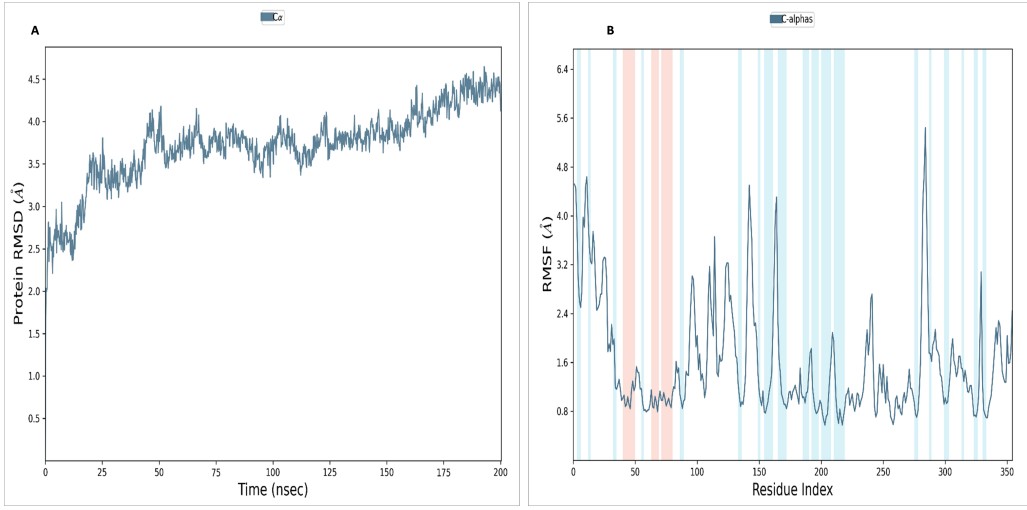

**Figure 7** (A) RMSD and (B) RMSF analysis of NS1 protein at 200 ns.

protein RMSD stabilized at 4.0 Å. The protein's RMSD showed oscillations mostly between 1.728 and 4.352 Å with an average RSMD of 3.806 ± 0.258 Å both in duplicates, indicating that the protein stayed in a stable structure during the simulation. In contrast, the RMSD trajectory of the L2 ligand began at 2.4 Å but increased significantly at two steps, first at 4.8 Å at 8 ns and second at 6.4 Å at 25 ns. Following this, the L2 ligand RMSD varied from 4.0 to 6.4 Å from 25 ns to 75 ns. After 75 ns, RMSD of ligand drastically fall down to 3.2 Å and remained at 3.2 up to 150 ns. After 150 ns RMSD increased to 6.4 Å at 180 ns and remained stable up to 200 ns. The ligand RMSD showed oscillations mostly between 0.855 and 6.59 Å with an average RSMD of 4.14 ± 0.971 Å both in duplicates. Overall, the ligand's RMSD indicates greater stability within the active binding site of the NS1 protein.

The RMSD of the NS1-L3 complex (Fig. 8B), which increased gradually to 7.2 Å at 25 ns from 2.408 Å at 0 ns. At 30 ns, the protein's RMSD trajectory decreased to 6.4 Å. The RMSD of protein fluctuated at 6.4 Å from 30 ns to 125 ns and further decreased to 5.6 Å and remained stable up to 200 ns. The RMSD of the protein fluctuated mostly between 5.6 and 6.4 Å, indicating that the protein stayed in a stable shape during the simulation. The protein's RMSD showed oscillations mostly between 2.408 and 7.249 Å with an average RSMD of 5.682 ± 0.474 Å both in duplicates. In contrast, the RMSD trajectory of the L3 ligand began at about 3.5 Å and increased quickly to 8.0 Å at 10 ns, with higher variations. Following this, the L3 ligand RMSD fluctuated between 8.0 and 7.0 Å between 25 to 150 ns before stabilizing at 7.0 Å from 160 ns to 200 ns. The L3 ligand RMSD was fluctuated between 2.67 and 8.54 Å with an average RSMD of 6.927 ± 0.673 Å both in duplicates. The ligand's RMSD generally indicates sustained stability within the NS1 protein's active binding region.

The NS1-L5 complex's RMSD (Fig. 8C) increased progressively to 4.5 Å at 15 ns and 4.8 Å at 25 ns from a starting point of about 3.0 Å at 0 ns. The protein's RMSD trajectory then varied from 25 to 100 ns, ranging from 4.8 to 4.2 Å. The protein RMSD stabilized

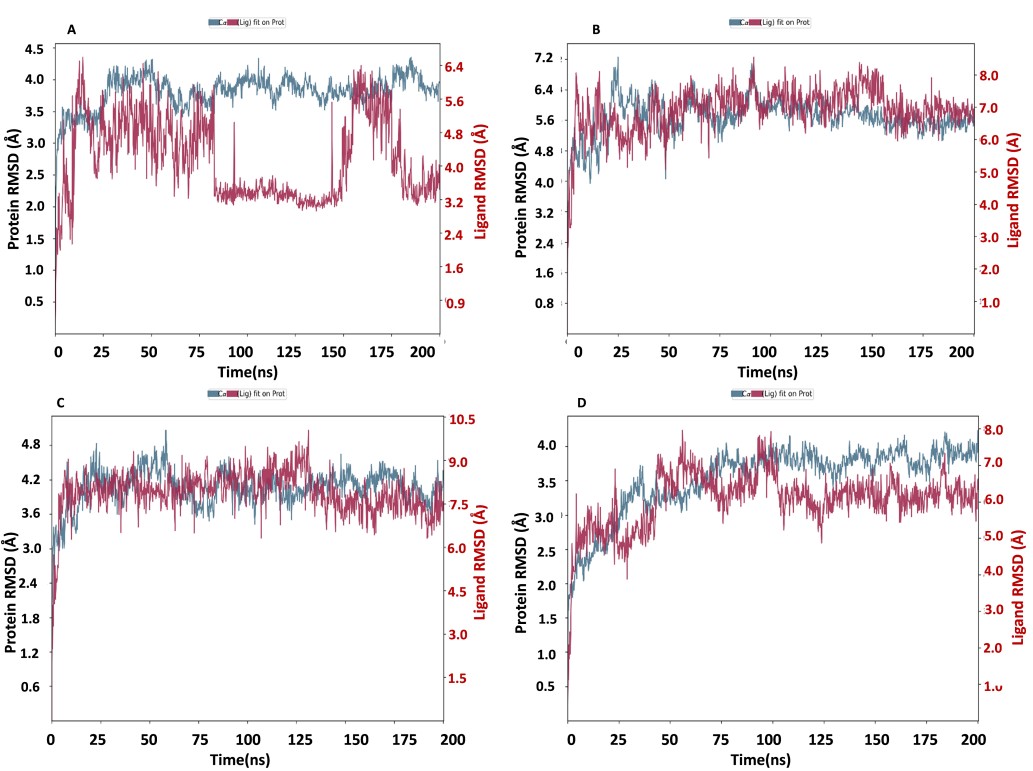

**Figure 8 RMSD trajectory of NS1-ligand complex at 200 ns of replica 1(A) NS1-L2 complex, (B) NS1-L3 complex, (C) NS1-L5 complex, and (D) NS1-dasabuvir complex.**

at about 4.2 Å up to 200 ns after 150 ns. When compared to other NS1 protein ligand complexes, the protein's RMSD fluctuated mostly between 3.0 and 4.8 Å, indicating that it kept a stable conformation throughout the simulation. The protein's RMSD showed oscillations mostly between 2.092 and 5.062 Å with an average RSMD of 4.059 ± 0.308 Å both in duplicates. In contrast, the RMSD trajectory of the L5 ligand began at about 4.0 Å but increased quickly to 9.0 Å at 20 ns with higher variations. Following this, the L5 ligand RMSD fluctuated from 7.5 Å to 9.0 Å from 15 ns to 125 ns and stabilized at 7.5 Å after 125 ns. The L5 ligand RMSD was fluctuated between 2.489 and 10.037 Å with an average RSMD of 7.897 ± 0.748 Å both in duplicates when compared to other protein-ligand complexes, the ligand's RMSD generally indicates increased stability inside the NS1 protein's active binding region.

Dasabuvir, an FDA-approved drug, was compared to the ligand and protein complex's RMSD in conjunction with the NS1 protein (Fig. 8D). Protein RMSD fluctuated narrowly, increasing from 1.8 Å at the beginning of the simulation to 3.5 Å at 40 ns. After 40 ns RMSD of protein increased to 4.0 Å at 75 ns and maintained stable at 4.0 Å up to 200 ns. The protein's RMSD showed oscillations mostly between 1.61 and 4.23 Å with an average RSMD of 3.531 ± 0.493 Å both in duplicates. In contrast, dasabuvir RMSD exhibited a fluctuation at active site of the NS1 up to 100 ns starting at 3.0 Å at the beginning of the simulation 8.0 Å at 100 ns. After 100 ns RMSD decreased sharply to 6.0 Å and maintained

stability up to 200 ns. The dasabuvir ligand RMSD was fluctuated between 1.119 and 7.956 Å with an average RSMD of 6.060 ± 0.775 Å both in duplicates. This demonstrated that dasabuvir, the ligand, remained stable at the NS1 protein's active site. In comparison RMSD of NS1-ligand complex, NS1-L2 complex showed larger fluctuations compared to other complex. The RMSD trajectory of NS1-ligand complex of replica 2 was represented in Fig. S1.

In addition, a third independent simulation was performed for each NS1–ligand complex using the same initial geometry but with randomized velocities from the Maxwell–Boltzmann distribution at 300 K to ensure statistical independence. This run was conducted to follow best MD practices for validating reproducibility. The RMSD profiles confirmed overall stability, showing different protein and ligand fluctuation patterns in the duplicates, with minor expected variations. These results support the robustness of the findings and confirm that the observed dynamics are dependent on specific initial conditions. RMSD trajectories from this run are provided in Fig. S2.

**RMSF of NS1 protein-ligand complex**

In Fig. 9, the RMSF map of the NS1 protein-ligand complex shows green bars indicating amino acids that are interacting with the ligand. The average RMSF for most amino acids in the RMSF plots for NS1-L2 (Fig. 9A), NS1-L3 (Fig. 9B), NS1-L5 (Fig. 9C), and NS1-dasabuvir (Fig. 9D) ligand complexes remains below 4.8 Å, 6.0 Å, 4.8 Å, and 5.0 Å, respectively. The RMSF larger than 8.0 Å of NS1-L3 and L5 complex fluctuated more from 1 to 15 amino acids at N-terminal. In both the complex, NS1 made contact with the ligand after 15th amino acid due to this interaction the RMSF value of protein reduced (*Roy et al., 2024*). The RMSF values for most of the amino acid residues of secondary structural element of NS1 involved in the interaction with the ligand were less than 2.0 Å. The amino acids present in non secondary structural element of NS1 protein involved in interaction with ligand showed larger fluctuations up to 4.8 Å for L2 and dasabuvir ligand and up to 3.0 Å for L3 and L5 ligand. The N and C-terminal area exhibited a particularly noticeable observation, which was characterized by a loop shape and resulted in large changes in RMSF (*Muthukumaran & Sankararamakrishnan, 2024*). When compared directly to the protein's unbound state (as shown in Fig. 7B), the NS1 protein-ligand complex had reduced RMSF values. This is likely due to the ligand binding in the loop region of the protein. This indicates that the complex exhibits reduced RMSF in comparison to the protein in its unbound form. It is important to mention that secondary structural elements, such as alpha helices and beta strands, generally show less variation compared to loop regions (shown by a white background) in all NS1-ligand complexes. The RMSF trajectory of NS1-L2 ligand complex was fluctuated between 0.642 and 4.956 Å with an average of 1.60 ± 0.73 Å, for NS1-L3 complex between 0.846 and 9.77 Å with an average of 2.011 ± 1.205 Å, for NS1-L5 complex between 0.699 and 8.608 Å with an average of 1.797 ± 0.979 Å, and similarly for NS1-dasabuvir complex between 0.539 and 5.32 Å with an average of 1.587 ± 0.820 Å. Similarly RMSF trajectory for NS1-ligand complex for replica 2 at 200ns was represented in Fig. S3.

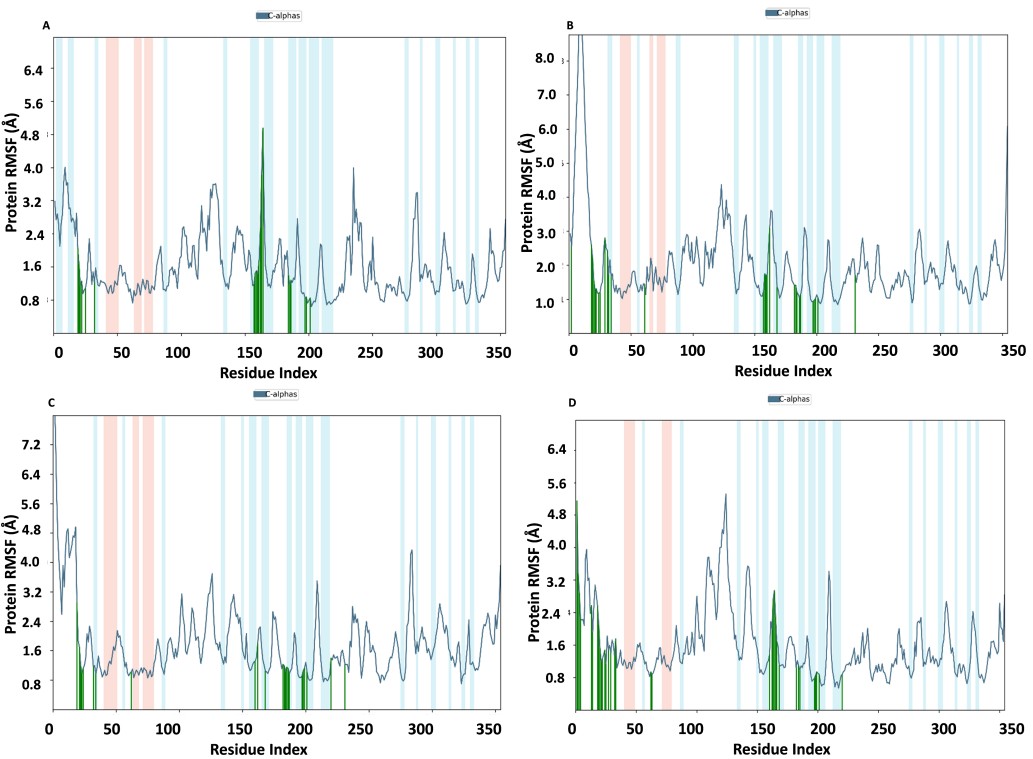

**Figure 9** RMSF trajectory of NS1-ligand complex at 200 ns of replica 1(A) NS1-L2 complex, (B) NS1-L3 complex, (C) NS1-L5 complex, and (D) NS1-dasabuvir complex.

In addition, a third independent simulation was performed for each NS1–ligand complex using the same initial geometry but with randomized velocities from the Maxwell–Boltzmann distribution at 300 K to ensure statistical independence. The RMSF profiles revealed deviations in fluctuation patterns compared to the duplicate runs, particularly in loop and surface-exposed regions of the protein. These variations are consistent with differences in initial velocity distributions and highlight the dynamic flexibility of certain residues. Despite these fluctuations, key binding site residues remained stable, supporting the structural robustness of the complexes. The RMSF plots from the third run are presented in Fig. S4.

Figure 10 displays the RMSF plot of the ligand for all NS1-ligand complexes. The mean RMSF results suggest that the displacement of each atom in the RMSF diagrams of the L2 ligand (Fig. 10A) was fluctuated between 1.206 and 4.723 Å with an average of 2.516 ± 0.931 Å, L3 ligand (Fig. 10B) was fluctuated between 1.062 and 2.248 Å with an average of 1.363 ± 0.286 Å, L5 ligand (Fig. 10C) was fluctuated between 1.043 and 3.2 Å with an average of 1.75 ± 0.637 Å, and dasabuvir ligand (Fig. 10D) was fluctuated between 0.964 and 3.149 Å with an average of 1.58 ± 0.60 Å. This indicates that the ligand consistently investigates several binding positions in order to find a more stable arrangement within the NS1 protein's binding region. Similarly, RMSF plot of all the ligands for all NS1-ligand complexes of replica 2 was represented in Fig. S5. Similarly, the RMSF plot of the ligand for

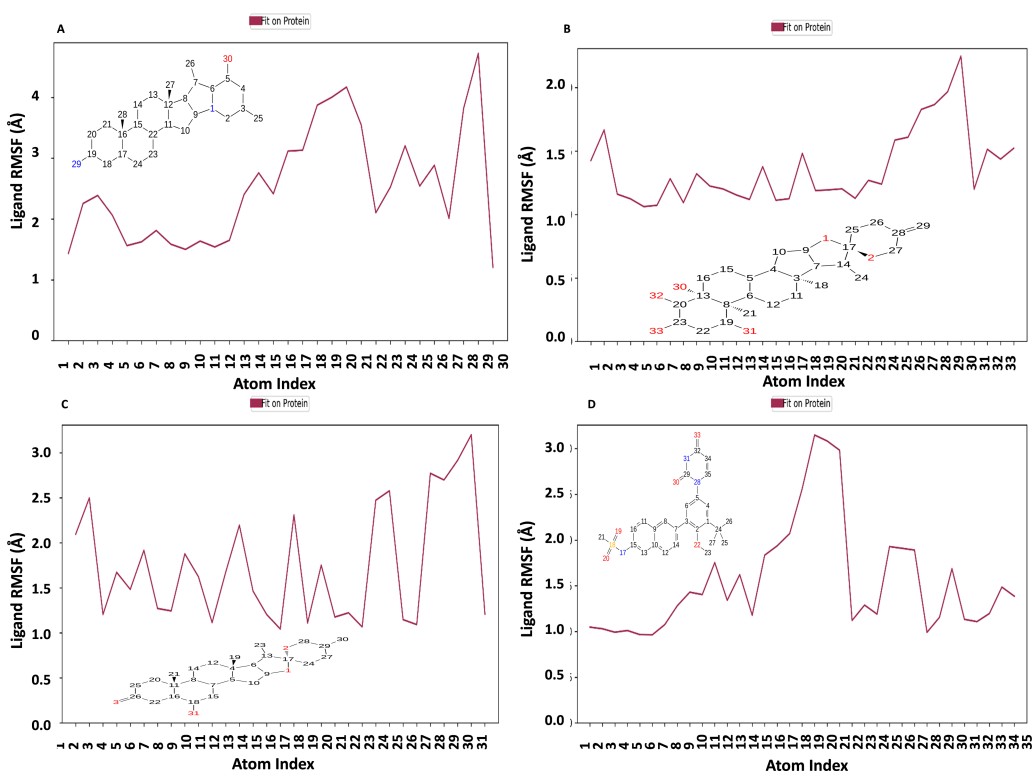

**Figure 10** Ligand RMSF plot of NS1-ligand complex at 200 ns for replica1 (A) L2 ligand, (B) L3 ligand, (C) L5 ligand, and (D) dasabuvir ligand.

**Table 11** Average value of RMSD and RMSF values of NS1 and NS1-ligand complex.

| Molecule | | Protein RMSD (Å) | Ligand RMSD (Å) | Protein RMSF (Å) | Ligand RMSF (Å) |
|---|---|---|---|---|---|
| NS1 | With the same initial velocity | 3.710 ± 0.490 | – | 1.60 ± 0.910 | – |
| | With a random initial velocity | 4.722 ± 0.491 | – | 1.454 ± 1.079 | – |
| NS1-L2 complex | With the same initial velocity | 3.805 ± 0.258 | 4.14 ± 0.971 | 1.60 ± 0.73 | 2.516 ± 0.931 |
| | With a random initial velocity | 4.288 ± 0.368 | 4.14 ± 0.971 | 1.60 ± 0.73 | 2.516 ± 0.931 |
| NS1-L3 complex | With the same initial velocity | 5.682 ± 0.494 | 6.927 ± 0.673 | 2.011 ± 1.205 | 1.365 ± 0.286 |
| | With a random initial velocity | 5.062 ± 0.601 | 8.307 ± 1.362 | 1.641 ± 0.771 | 2.119 ± 0.687 |
| NS1-L5 complex | With the same initial velocity | 4.059 ± 0.308 | 7.807 ± 0.748 | 1.707 ± 0.979 | 1.75 ± 0.637 |
| | With a random initial velocity | 4.300 ± 0.527 | 3.142 ± 0.662 | 1.890 ± 1.558 | 1.473 ± 0.287 |
| NS1-Dasabuvir Complex | With the same initial velocity | 3.531 ± 0.493 | 6.060 ± 0.775 | 1.587 ± 0.82 | 1.54 ± 0.60 |
| | With a random initial velocity | 4.406 ± 0.386 | 2.897 ± 0.603 | 1.651 ± 1.067 | 1.606 ± 0.524 |

all NS1-ligand complexes with random velocities was represented in Fig. S6. The summary of RMSD of ligand-protein complex and RMSF of protein and ligand are represented in Table 11.

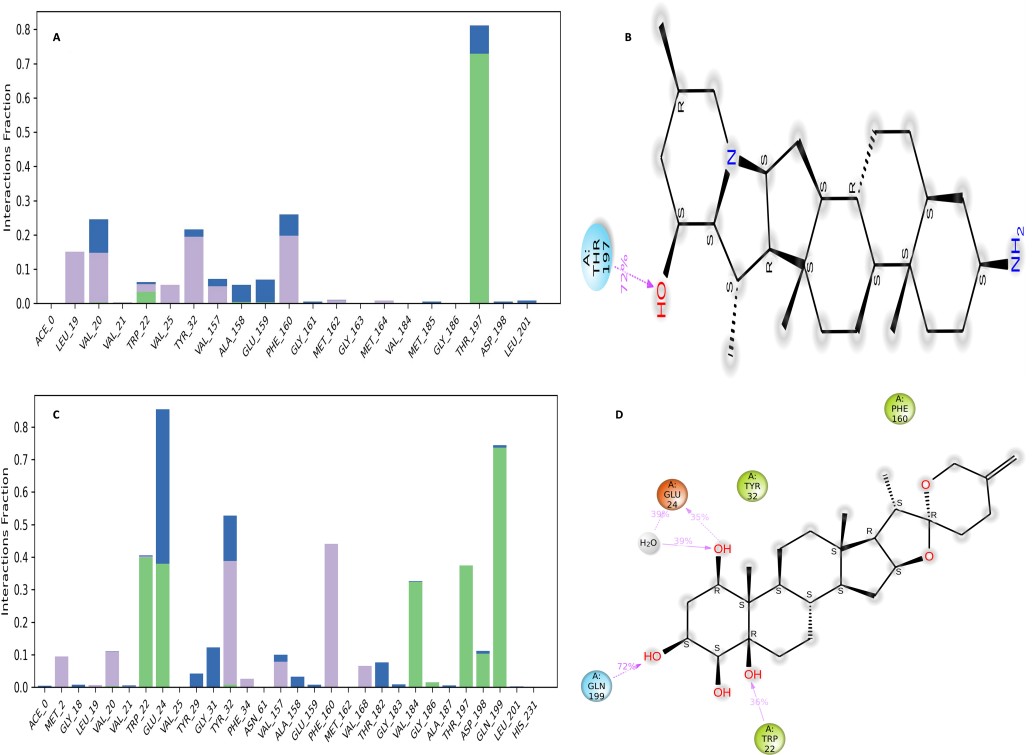

**Figure 11** NS1-ligand interaction map of replica1: (A) NS1-L2 complex, (C) NS1-L3 complex. NS1-ligand contact for more than 30% simulation time (B) NS1-L2 complex, (D) NS1-L3 complex.

## Interaction diagram of NS1 protein-ligand complex

The NS1 protein-ligand complex during the 200 ns MD simulation is depicted in Figs. 11 and 12, where four distinct forms of protein-ligand contacts are identified: hydrophobic, ionic, hydrogen bonds, and water bridges. During the course of the analysis, these stacked bar charts are standardized; higher interaction fractions suggest longer-lasting interactions. According to the interaction map of the NS1-L2 complex (Fig. 11A), L2 formed less hydrogen bonds with amino acids of NS1 protein during the MD simulation. The specific amino acids involved in the hydrogen bond were Trp22 and Thr197. The L2 ligand exhibits a higher number of interactions with the NS1 protein through hydrophobic interactions and water bridges compared to hydrogen bonds. Figure 11B provides a concise overview of the protein-ligand contacts illustrating the many interactions and contacts (such as hydrophobic, ionic, H-bonds, and water bridges) between the L2 ligand and NS1 protein, which occurred for more than 30% of the simulation duration. During the simulation, the amino acids Thr197 maintained contact time of 72% with L2 ligand. This indicates L2 ligand formed less contact with NS1 protein during simulation. This may be due to larger fluctuation of ligand RMSD as indicated by ligand RMSD diagram.

According to the interaction diagram (Fig. 11C) of the NS1-L3 complex, L3 formed typical hydrogen bonds with amino acids of NS1 protein during the MD simulation. The amino acids involved in these bonds were Trp22, Glu24, Val184, Gly186, Thr197, Asp198,

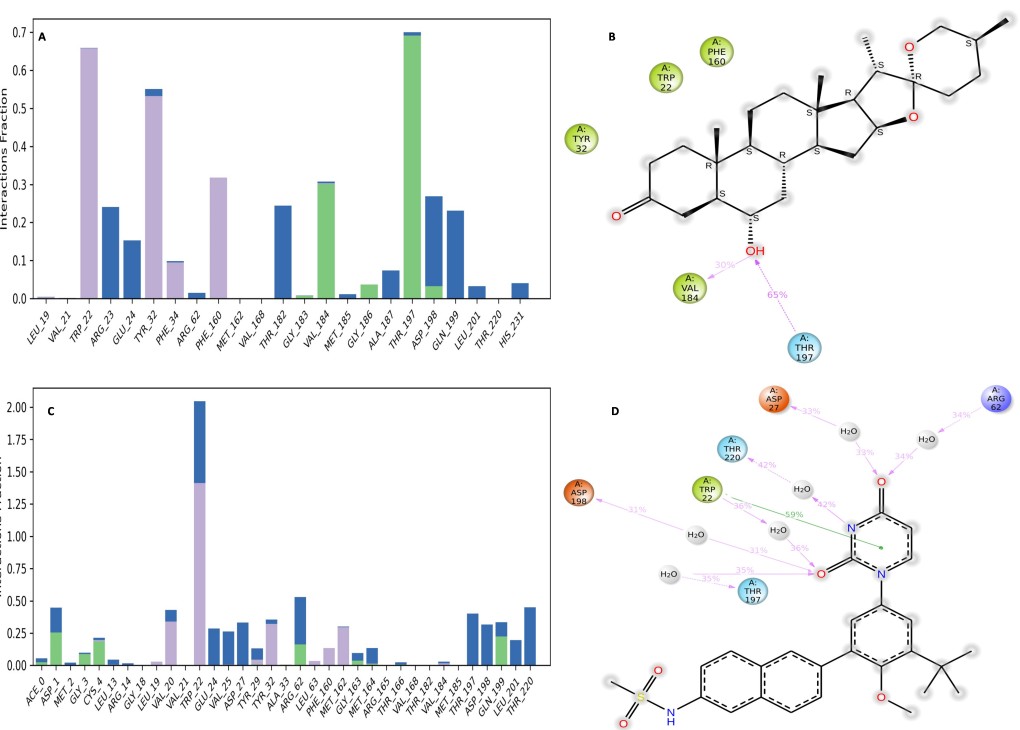

**Figure 12** NS1-ligand interaction map of replica1 (A) NS1-L5 complex, and (C) NS1-dasabuvir complex and NS1-ligand contact for more than 30% simulation time (B) NS1-L5 complex, and (D) NS1-dasabuvir complex.

and Gln199. The L3 ligand also exhibits a greater number of interactions with the NS1 protein through water bridges and hydrophobic interactions similar to hydrogen bonds. Figure 11D provides a summary of the protein-ligand contact showing the interactions and contacts that occurred for more than 30% of the simulation duration are included. The amino acids Trp22, Glu24, Tyr32, Phe160, and Gln199 maintained contact times of 22%, 39%, 40%,45%, and 72%, respectively, during the simulation.

The NS1-L5 complex interaction diagram (Fig. 12A) shows that during the MD simulation, L5 forms hydrogen bonds with the amino acids of the NS1 protein. These amino acids include Gly183, Val184, Gly186, Thr197, and Asp198. Figure 12B provides a summary of the protein-ligand contact illustrating the interactions and contacts between the L5 ligand and NS1 protein that occurred for more than 30% of the simulation duration. The amino acids Trp22, Tyr32, Phe160, Val184, and Thr197 maintained contact times of 60%, 55%, 32%, 30%, and 65%, respectively, during the simulation.

In the interaction diagram (Fig. 12C) of the NS1-dasabuvir complex, dasabuvir formed traditional hydrogen bonds with several amino acids and the ligand during the MD simulation. These amino acids include Asp1, Gly3, Cys4, Arg62, Gly163, Met164, Gln199, and also a large number of water bridges. Figure 12D provides a concise overview of the protein-ligand contact between the dasabuvir ligand and NS1 protein, specifically focusing on interactions that occurred for more than 30% of the simulation duration. The amino

acids Trp22, Asp27, Arg62, Thr197, Asp198, and Thr220 retained 59%, 33%, 34%, 35%, 31%, and 42% of their contact time, respectively, throughout the simulation. Similarly, the interaction diagram of all the NS1-ligand complexes of replica 2 was represented in Fig. S7.

In addition, a third independent simulation was conducted using randomized initial velocities to assess the consistency of protein–ligand interactions under varied initial conditions. While the core set of interacting residues remained largely conserved across all three runs, variations were observed in the contact duration and frequency of specific interactions. In particular, key residues within the binding pocket maintained consistent interactions with the ligand, but the percentage of simulation time during which these interactions were present differed between the duplicates and the third run. These differences arise from slight shifts in ligand orientation and protein side-chain dynamics induced by the altered initial velocities. Such variations are expected in MD simulations and reflect the natural flexibility of molecular systems, reinforcing the dynamic nature of ligand binding while confirming the overall stability and reliability of the binding mode. The interaction profiles from the third run are provided in Fig. S8.

Further, to understand the stability of binding modes between ligand and NS1 protein in MD simulation, contact analysis was conducted between ligand and NS1 protein for the initial structure (0 ns), intermediate structure (100 ns), and final structure (200 ns) of MD simulation and represented in Figs. 13 and 14. The ligand-protein interaction distances evolved differently across the simulations, revealing dynamic shifts in binding stability. In the L2 complex (Figs. 13A, 13B and 13C), the hydrogen bond with Thr197 remained relatively stable for around 72% of simulation time with the O-group of the ligand, with a slight fluctuation from 2.18 Å at 0 ns, decreasing to 1.88 Å at 100 ns, and returning to 2.18 Å at 200 ns. This suggests that the ligand maintained a strong and consistent interaction within the binding pocket. Similarly, the L3 complex (Figs. 13D, 13E and 13F) amino acid Gln199 formed stable hydrogen bond interaction for around 72% of simulation time with the O-group of the ligand, with a slight fluctuation from 2.01 Å at 0 ns, decreasing to 1.85 Å at 100 ns, and returning to 2.01 Å at 200 ns. This suggests that the ligand maintained a strong and consistent interaction within the binding pocket. The L5 complex (Figs. 14A, 14B and 14C) exhibited a relatively stable interaction with Thr197 for around 65% of simulation time with the O-group of the ligand, where the bond distance was 1.97 Å at 0 ns, slightly increased to 2.01 Å at 100 ns, and then decreased to 1.97 Å at 200 ns. These minor fluctuations indicate that the ligand remained engaged with the binding site but experienced small positional adjustments during the simulation. Meanwhile, the Dasabuvir complex (Figs. 14D, 14E and 14F) showed a larger variation. For example, Met162 formed a hydrogen bond with a ligand with a distance of 1.95 Å at 0 ns to 4.57 Å at 100 ns and formed an alkyl bond with a ligand and further to 4.75 Å at 200 ns. This suggests the ligand displacement within the binding pocket.

## Interaction timeline diagram of the NS1 protein-ligand complex

Figures 15 and 16 depict the quantity of contact ligands (shown in the top panel) that interact with the NS1 protein. The bottom panel of the figure displays a timeline diagram illustrating the interaction between the NS1 protein and the ligand complex. During the
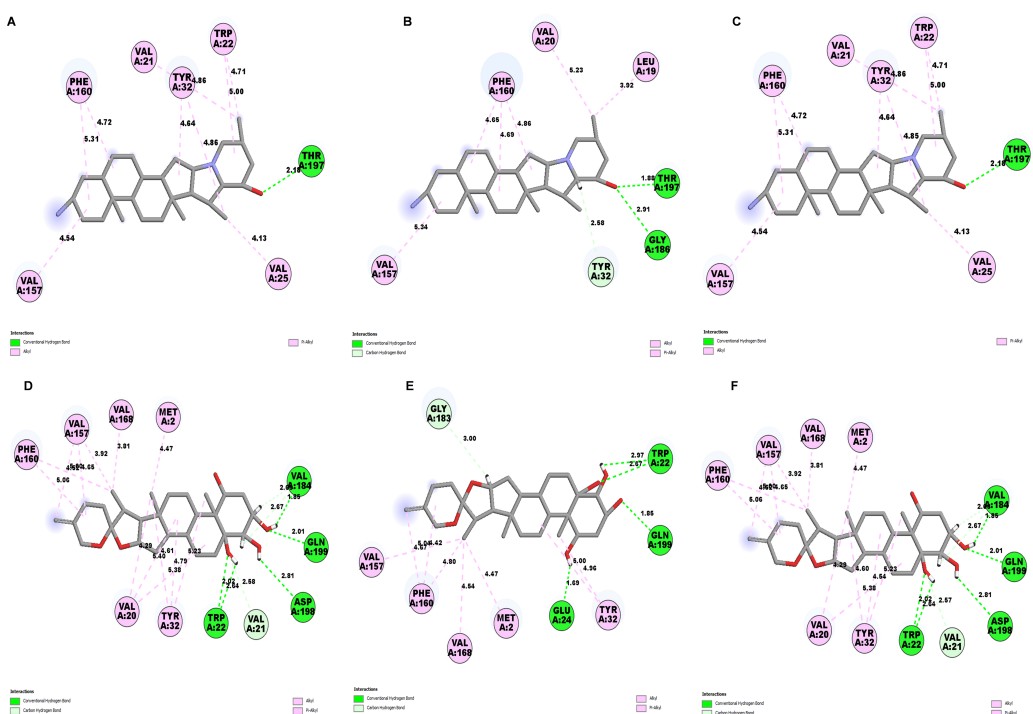

**Figure 13** NS1-ligand contact analysis 2D interaction diagram: (A, B & C) NS1-L2 complex at 0 ns, 100 ns and 200 ns, (D, E, & F) NS1-L3 complex at 0 ns, 100 ns and 200 ns, (G, H, & I).

modeling of the NS1 protein-L2 complex (Fig. 15A), the protein and ligand had a varying number of unique interactions, ranging from one to six amino acids. Several residues, including Thr197, Phe160, Tyr32, Val20, and Leu19, exhibited multiple interactions with the ligand, as shown by an orange color. The ligand underwent a shift from its initial binding location throughout the simulation time, but maintained stable contact with Thr197 and Phe160 amino acids. This is consistent with the information presented in Fig. 8A, which demonstrates variations in the ligand's RMSD and an unsteady binding position. During the simulation, the NS1 protein-L3 complex (Fig. 15B) exhibited a range of three to 10 unique interactions between the protein and ligand, involving amino acids. The ligand made multiple contacts with certain residues, namely Gln100, Thr197, Phe160, Tyr32, Glu24, and Trp22 as evidenced by an orange tone. These findings indicate that the ligand underwent a change in its initial binding location during the MD simulation but maintained constant binding with Gln199 and Glu24. This is consistent with Fig. 8B, which demonstrates ligands' RMSD is stable and maintains a stable binding conformation. During the simulation, the NS1 protein-L5 complex (Fig. 16A) exhibited a range of three to 10 amino acids with unique interactions between the protein and ligand. Residues Thr197, Val184, Thr182, Phe160, Tyr32, and Trp22 exhibited several interactions with the ligand, as evidenced by an orange tone. These residues maintained a consistent contact throughout the MD simulation, as depicted in Fig. 8C, which also showed reduced variations in the ligand's RMSD. In the case of the NS1 protein-dasabuvir complex (Fig. 16B), the simulation

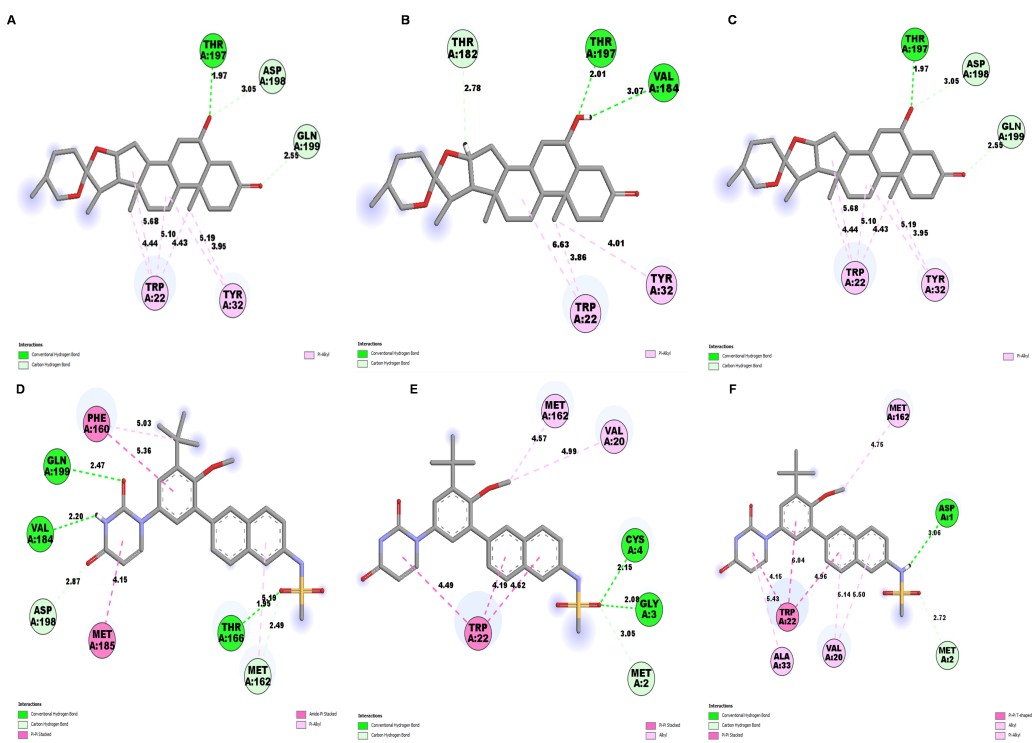

**Figure 14  NS1-ligand contact analysis 2D interaction diagram (A, B & C) NS1-L5 complex at 0 ns, 100 ns and 200 ns, (D, E, & F) NS1-dasabuvir complex at 0 ns, 100 ns and 200 ns.**

revealed that there were between four and 16 unique interactions between the protein and ligand involving amino acids. Residues Gln199, Arg62, Trp22, and Val20 exhibited several interactions with the ligand, as indicated by an orange shade. The timeline figure illustrates the evolving interactions between the NS1 protein and the ligand during a 200 ns MD simulation. Similarly, the interaction timeline diagram of all the NS1-ligand complexes of replica 2 was represented in Fig. S9.

## Binding poses of the NS1 protein-ligand complex

Throughout the 200 nanosecond MD simulation, the ligands experienced conformational alterations within the active binding pocket of the NS1 protein. These changes occurred as the ligands sought to stabilize themselves by locating more appropriate binding positions. This is apparent in the ligand RMSD trajectories depicted in Figs. 8A to 8D. The ligands aimed to achieve stability by interacting with various amino acid residues over a period of time, as demonstrated by the protein-ligand interaction timeline depicted in Fig. 17 for NS1 protein-L2 complex, which attach to the active binding pocket of the NS1 protein at various time periods, namely 0 ns, 40 ns, 80 ns, 120 ns, 160 ns, and 200 ns. Similarly, binding poses for other NS1 protein-ligand complexes were depicted in supplementary information in Fig. S10 for the NS1 –L3 complex, Fig. S11 for the NS1 –L5 complex, and Fig. S12 for the NS1 –dasabuvir complex at various time periods, namely 0 ns, 40 ns, 80 ns, 120 ns, 160 ns, and 200 ns. Binding poses of replica2 for NS1-L2 complex, NS1-L3

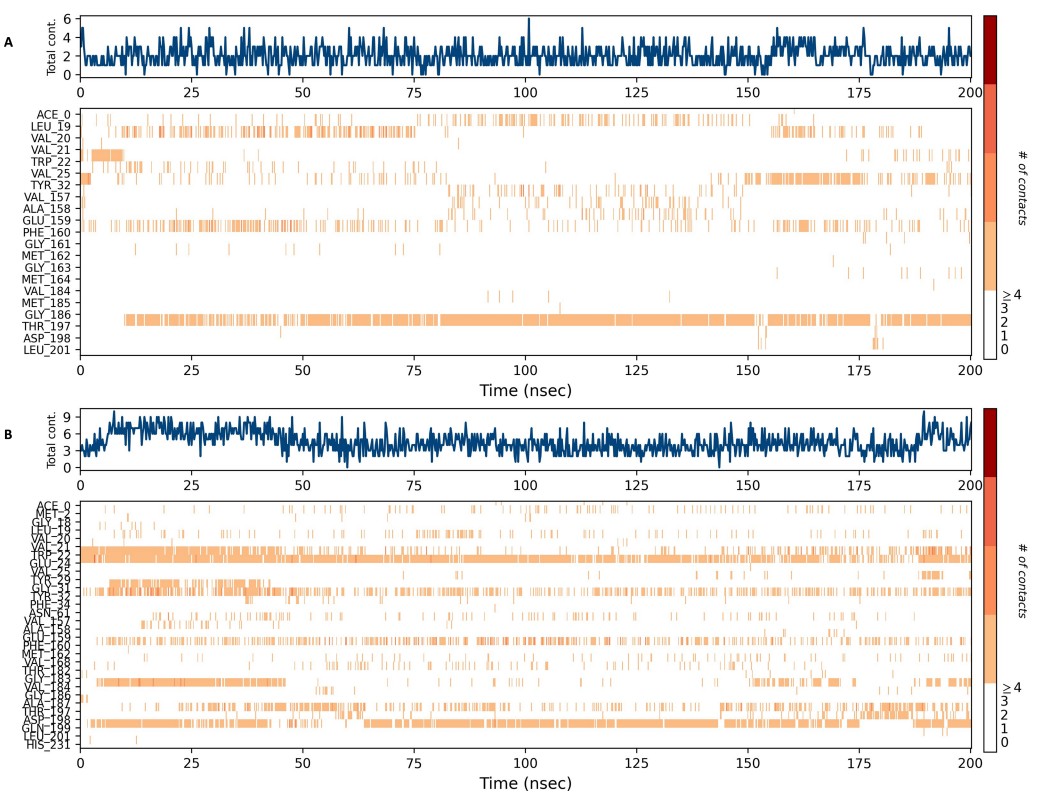

**Figure 15** NS1 protein-ligand interaction timeline (A) NS1-L2 complex, and (B) NS1-L3 complex.

complex, NS1-L5 complex, and NS1-Dasabuvir complex are represented in Figs. S13, S14, S15, and S16 respectively.

## Binding free energy analysis of NS1 protein-Ligand complex

In this study, we employed the prime MM-GBSA method to estimate the binding free energies of ligand-protein complexes during molecular dynamics simulations. We evaluated energy components such as electrostatic interactions, covalent contributions, hydrogen bonding, van der Waals forces, self-contact energies, lipophilic interactions, and solvation effects. These factors were combined to calculate the binding free energy ($\Delta G_{bind}$) in kcal/mol, offering insights into the strength and favourability of ligand-protein interactions, a standard practice in drug discovery and molecular dynamics research. Using the MD simulation trajectory, we calculated the binding free energy and other contributing energies *via* MM-GBSA for the ligand complexes. As shown in Table 12, the average binding free energy of the NS1 protein with the standard antiviral drug dasabuvir complex ($-87.68 \pm 4.31$ kcal/mol) is slightly higher than that of the L3 ($-77.224.713$ kcal/mol). With respect to MM-GBSA results, similar to FDA-approved drug dasabuvir, L3, L2 and L5 ligands can be considered as probable antivirals for KFDV.

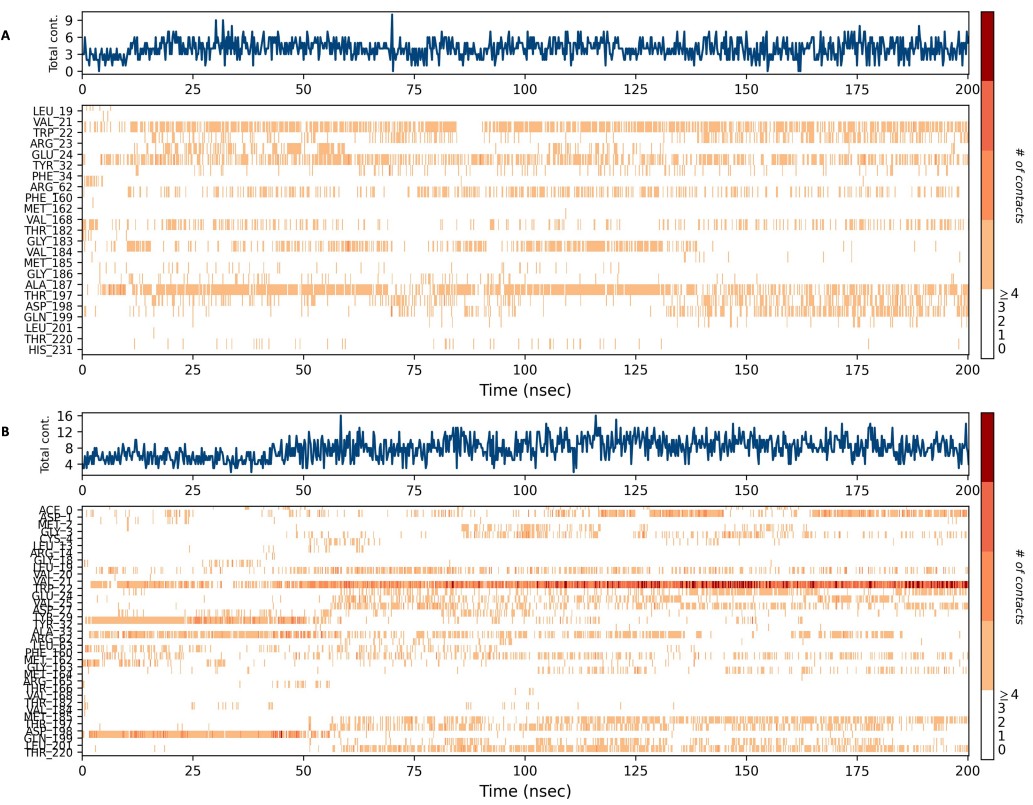

**Figure 16** NS1 protein-ligand interaction timeline (A) NS1-L5 complex, and (B) NS1-dasabuvir complex.

**Table 12 Binding free energy components calculated by MM-GBSA for L3, L5, L6, and dasabuvir complex.**

| Energies (kcal/mol) | L2 complex | L3 complex | L5 complex | Dasabuvir complex |
|---|---|---|---|---|
| ΔGbind | −62.97 ± 4.00 | −77.22 ± 4.71 | −62.07 ± 2.88 | −87.68 ± 4.31 |
| ΔGbindLipo | −30.10 ± 1.67 | −30.04 ± 1.67 | −27.37 ± 1.56 | −27.66 ± 1.09 |
| ΔGbindvdW | −45.66 ± 2.73 | −56.31 ± 2.81 | −46.35 ± 1.95 | −67.20 ± 2.47 |
| ΔGbindCoulomb | −10.81 ± 1.60 | −19.24 ± 4.35 | −11.49 ± 1.72 | −17.47 ± 3.31 |
| ΔGbindHbond | −0.53 ± 0.05 | −2.09 ± 0.32 | −0.47 ± 0.19 | −0.65 ± 0.24 |
| ΔGbindSolvGB | 21.99 ± 1.86 | 28.49 ± 2.36 | 23.36 ± 1.57 | 32.35 ± 1.92 |
| ΔGbindCovalent | 2.13 ± 0.78 | 1.98 ± 1.80 | 0.26 ± 0.74 | 0.84 ± 1.46 |

## DISCUSSION

Given the limited efficacy of the vaccine in combating KFDV, the public health response is weakened, making it less effective in addressing the situation (*Bhatia et al., 2023*; *Srikanth et al., 2023*; *Kasabi et al., 2013*; *Kiran et al., 2015*). The rising number of KFDV cases and the expanding endemic areas in India highlight the need for an interdisciplinary approach to develop antiviral drug candidates for KFDV (*Malik et al., 2023*). This study aimed to inhibit the function of the NS1 protein, which is a multifunctional protein involved in viral

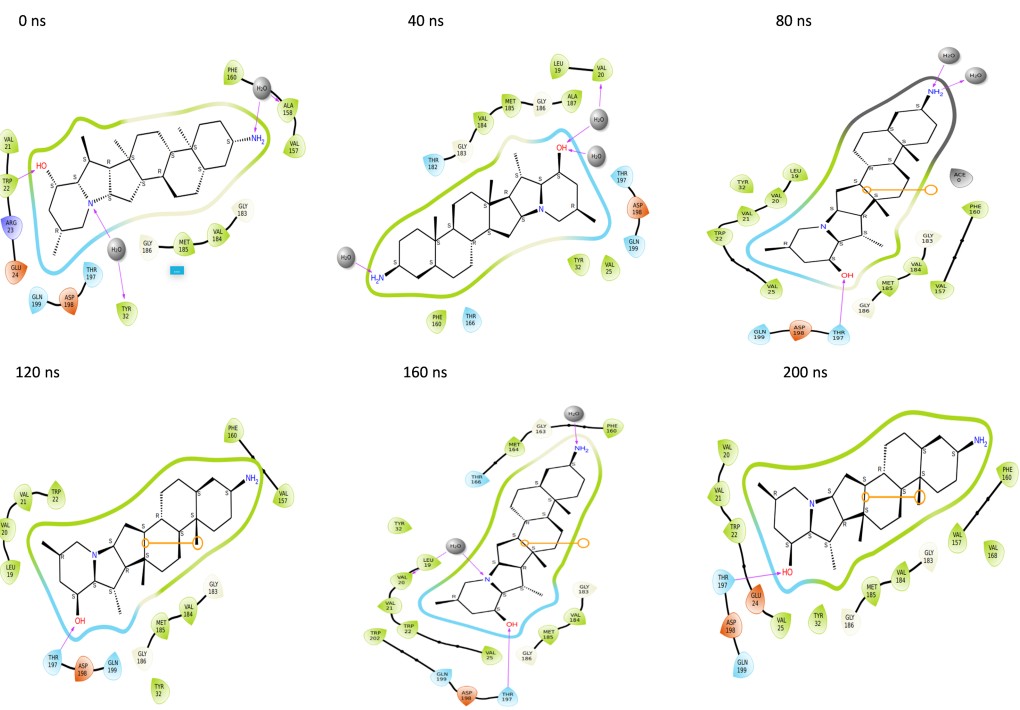

**Figure 17** Binding poses of L2 ligand at the active site of NS1 protein at different time intervals during MD simulation of replica1.

replication, immune evasion, and host cell interactions, by identifying a novel and effective antiviral drug candidate (*Denolly et al., 2023*; *Banjan et al., 2024*; *Chew et al., 2024*) against KFDV using virtual screening of compounds from the IMPPAT database. A similar study for finding a potential antiviral agent from phytochemicals has been reported. Phytochemicals as potential inhibitors for Dengue virus NS2D-NS3 protease from the PubChem database (*Purohit et al., 2024*), for capsid protein (*Khan, Hasan & Hasan, 2024*), phytochemicals from *Piper caninum* as against Dengue NS2B-NS3 protease inhibitor (*Mustafa et al., 2024*), phytochemicals of *C. wightii* against Dengue target NS5 and envelope protein (*Jain, Singh & Kumari, 2024*), and traditionally used medicinal plants against Dengue virus (*Islam et al., 2024*).

In this study, we obtained the NS1 protein sequence (778–1,130 of 353 amino acids) from the UniProt database (UniProt ID: D7RF80.1) and conducted a sequence analysis. Sequence analysis of NS1 revealed that the protein has homology with Zika virus and West Nile virus. The lack of a 3D structure for the NS1 protein necessitated the use of Robetta, Swiss Model, and I-Tasser MTD servers to establish its 3D structure. The model developed from I-Tasser was chosen based on the structural validation procedure and then underwent energy minimization utilizing the YASARA energy minimization server. The minimized 3D structure of the NS1 protein exhibited an increase in the ERRAT score from 86.09 to 94.37, indicating that the minimized model is of high quality and can be further utilized in the process of drug screening process. Prediction of protein-ligand binding sites is crucial for annotating protein activity and discovering drugs. Various techniques for active site

prediction include sequence-based, structure-based, and template-based methods. Among these, structure-based prediction is one of the methods for predicting ligand binding sites. This is achieved by finding interior voids and surface pockets of the proteins, which are prominent concave regions of proteins that are frequently associated with binding events (*Adebiyi & Obagbuwa, 2024*).

The active site analysis of the NS1 protein of KFDV identified key residues potentially involved in protein-protein or ligand interactions. The selected binding pocket includes Asp1, Met2, Gly3, Cys4, Ala5, Ile6, Asp7, Ala8, Asn9, Arg10, Arg14, Gly16, Glu17, Gly18, Leu19, Val20, Val21, Trp22, Arg23, Glu24, Asp27, Tyr32, Phe160, Gly161, Met162, Thr166, Gly183, Val184, Met185, Gly186, Ala188, Val189, Lys190, Ser191, His193, Ala194, Val195, Thr197, Asp198, Gln199, and Trp202. Comparatively, the dengue virus DENV NS1 protein has well-characterized functional sites, including its $\beta$-roll (residues 1–29), wing domain (30–180), and $\beta$-ladder (181–352) (*Shu et al., 2022*). In DENV NS1, key residues such as Cys4, Asn130, Asn207, and Trp115 play crucial roles in dimer stability, glycosylation, and host immune interactions. Notably, Cys4, which is also present in the KFDV NS1 active site, is conserved across flaviviruses and is essential for maintaining NS1's structural integrity. Additionally, residues within the wing domain (*e.g.*, Tyr32) and $\beta$-ladder (*e.g.*, Gly183, Val184, and Met185) in KFDV NS1 align with functionally significant regions in DENV NS1, suggesting a conserved role in structural stability and host interactions (*Yadav et al., 2019*). Although NS1 proteins across flaviviruses share structural and functional similarities, sequence variations within their active sites may influence interactions with host factors and immune responses. The identification of a well-defined binding pocket in KFDV NS1 strengthens its potential as a target for antiviral development, particularly in disrupting its immune evasion mechanisms (*Chen, Lai & Yeh, 2018*). The coordinates of the NS1 protein (X 59.47, Y 57.19, Z 55.68 in Å) were calculated using the Biovia Discovery Studio. These coordinates were then utilized to discover ligands based on virtual screening of the IMPPAT database. The compounds from the IMPPAT database were filtered by Lipinski RO5.

A total of 11,530 compounds from the IMPPAT database were considered for virtual screening using PyRx 0.8 with the standard FDA-approved antiviral medicines. Screening of drugs was carried out in three levels, primary screening with an exhaustiveness of 12 for 11,530 compounds. Based on the minimum binding energy value top 10% (1,150) compounds underwent secondary screening with an exhaustiveness of 100. Based on the binding affinity value, the top 10% (115) compounds were taken to tertiary screening using AutoDock 4.0 implemented in PyRx 0.8 and compared with FDA-approved drugs. Out of the FDA-approved drugs, Dasabuvir was selected as a reference drug due to its established efficacy as an FDA-approved non-nucleoside inhibitor targeting the NS5B RNA-dependent RNA polymerase of the Hepatitis C virus (HCV). By binding to an allosteric site on the NS5B polymerase, Dasabuvir effectively inhibits viral RNA replication, thereby serving as a critical component in HCV treatment regimens (*Trivella, Gutierrez & Martin, 2015*). KFDV is a member of the Flaviviridae family, sharing structural and functional similarities with HCV. Notably, KFDV encodes several non-structural proteins, including NS1, which plays a pivotal role in viral replication and immune evasion. The NS1
protein is involved in modulating host immune responses and is essential for the virus's life cycle (*Yadav et al., 2019*). Given the functional similarities between HCV's NS5B polymerase and KFDV's NS1 protein in terms of their contributions to viral replication, Dasabuvir was chosen as a reference compound. This selection facilitates the evaluation of potential inhibitors targeting KFDV's NS1 protein by providing a benchmark for antiviral efficacy. By comparing novel compounds to Dasabuvir, we aim to identify candidates that can disrupt KFDV replication through mechanisms analogous to those employed against HCV.

A total of 15 compounds with minimum binding energy value and the FDA-approved drug dasabuvir were considered for toxicity prediction. Fifteen compounds were analyzed for toxicity using ProTox II and compared to FDA-approved antiviral drugs, Dasabuvir as a standard. The compounds L2, L3, L5, L6, L7, and L9, and Dasabuvir were selected for additional analysis in the field of ADME using the Swiss ADME server. In order to determine the suitability of these compounds as drugs, an ADMET analysis of the identified molecule was conducted, using dasabuvir as a reference. A similar approach was carried out by *Ejeh et al. (2021)* for Hepatitis C virus NS5B polymerase, and by *Shakya (2019)* for anti-HCV targets by considering dasabuvir as a reference compound. All of the chosen compounds are considered to satisfy Lipinski's criterion, having moderate water solubility, high GI absorption, and good synthetic accessibility compared to the standard FDA-approved dasabuvir drug. Therefore, the compounds L2, L3, L5 and L6 were selected for interaction analysis using Biovia Discovery Studio. Following the completion of ADMET, molecular docking was conducted. In summary, the results of the molecular docking analysis in terms of binding energy and interaction analysis between NS1 and four ligands, including dasabuvir, revealed that phytochemicals exhibited better binding affinity compared to the standard FDA-approved drug dasabuvir. The identified compounds exhibited equivalent or greater traditional hydrogen bonding compared to the reference molecule, suggesting a more effective contact with higher affinity (*Madushanka et al., 2023*; *Chagaleti et al., 2024*).

The phytochemical solanogantamine (IMPHY010294-L2), a steroidal alkaloid derived from *Solanum giganteum*, exhibited a binding affinity of −9.34 kcal/mol. Although there are no prior reports of solanogantamine being used as an antiviral agent, extracts of *Solanum giganteum* have demonstrated anti-inflammatory, antibacterial, antifungal, anticancer, and neuroprotective properties (*Shende & Kakadiya, 2022*; *Shende & Kakadiya, 2023*). Similarly, ranmogenin A (IMPHY001281-L3), a triterpenoid from *Rohdea nepalensis* (*Tupistra aurantiaca*), exhibited a binding affinity of −9.12 kcal/mol. While ranmogenin A has been reported for its inhibitory effects on human gastric tumor cells (*Pan et al., 2003*) and antifungal activity (*Wu et al.,2007*), its antiviral potential remains unexplored. Likewise, solagenin (IMPHY011162-L5), a triterpenoid from *Solanum asperolanatum*, exhibited a binding affinity of −9.08 kcal/mol. Although there are no specific reports on the medicinal application of solagenin, *S. asperolanatum* has been traditionally used to treat ailments such as diarrhea, stomach pain, whooping cough, diabetes, malaria, and ulcers (*Mwambo & Chuba, 2024*). The strong binding affinities observed in this study suggest that these phytochemicals may also possess antiviral potential against KFDV. However, further *in vitro* and *in vivo* investigations are necessary to validate their efficacy as antiviral agents.

Docking and molecular dynamics (MD) simulations are complementary approaches for studying protein-ligand interactions, offering distinct insights into binding mechanisms. Docking provides a static snapshot, predicting the most favorable binding pose based on shape complementarity and interaction energy. However, it does not fully account for protein and ligand flexibility beyond limited side-chain adjustments. In contrast, MD simulations capture the dynamic behavior of the complex over time, incorporating conformational changes, solvation effects, and thermal fluctuations. To ensure the stability of the protein-ligand complex, molecular docking must be validated through MD simulations for a minimum of 100 nanoseconds (*Akash et al., 2024*; *Roy et al., 2024*). In our study, while docking identified key interactions, MD simulations revealed that some contacts were lost or altered, while new interactions emerged due to conformational flexibility. This underscores the necessity of MD simulations in refining docking predictions and assessing binding stability under physiological conditions.

The docking complex structure for the NS1 protein-ligand L2, L3, L5 and dasabuvir complex was subjected to MD simulation to evaluate the stability of the compounds within the protein's binding region. The simulated trajectories lasting 200 nanoseconds, which encompassed measurements of RMSD, RMSF, protein-ligand contact mapping, and ligand characteristics (*Mohamed et al., 2024*), were subjected to analysis using the Desmond software. All the NS1-ligand complexes displayed stable conformations in the present investigation except the NS1-L2 complex. During the 200 ns simulation, the ligands in the protein-ligand complexes made efforts to find various binding positions while ensuring stability within the specified active site. The observed ligand-protein interaction distances suggest varying degrees of binding stability among the different ligand complexes. In the L2 and L3 complexes, the hydrogen bonds with Thr197 and Gln199, respectively, exhibited minimal fluctuations in bond distances. These interactions remained within a narrow range, indicating that the ligands maintained a strong and consistent binding within the active site throughout the simulation. The L5 complex showed a slightly greater variation in its interaction with Thr197, but the changes were minor, suggesting that while the ligand experienced some positional adjustments, it largely remained within the binding site. In contrast, the Dasabuvir complex demonstrated significant variation in its interaction with Met162, where the initial hydrogen bond was lost over time, transitioning into an alkyl interaction with an increased bond distance. This shift suggests that Dasabuvir underwent partial displacement from its initial binding position, potentially reducing its overall binding stability. Such behavior could indicate a weaker binding affinity or an alternative binding mode that may not be as effective in maintaining strong interactions over extended simulation periods. These findings emphasize the role of molecular dynamics in assessing ligand stability beyond static docking results. Previous studies have revealed similar findings, indicating that ligands actively search for stable binding sites during MD simulations (*Achappa et al., 2024*; *Wu et al., 2024*; *Handa et al., 2024*).

In addition, an assessment of the RMSF was performed for both the protein and the ligand using an all-atom MD simulation. The RMSF values of terminal amino acids were often greater than those of other residues, with the optimal values being below 4 Å. The amino acid residues that interact with the protein have RMSF values below 3 Å. During

the 200 nanosecond MD simulation, all ligands consistently maintained substantial connections with the protein. The observation was made regarding the association between ligand RMSD, ligand RMSF, and protein-ligand interactions. In addition, the MM-GBSA method was used to calculate the binding free energies of the ligand-protein complexes in MD simulations. The NS1 protein exhibits an average binding free energy of $-62.97 \pm 4.0$ kcal/mol with the L2 complex, $-77.22 \pm 4.21$ kcal/mol with the L3 complex, $-62.07 \pm 2.88$ kcal/mol with the L5 complex, and $-87.68 \pm 4.31$ kcal/mol with the dasabuvir complex. This suggests that the L3 and L2 ligands have a comparable binding affinity to dasabuvir. These findings indicate that the two ligands, L3 and L2, can be proposed as probable inhibitors for the NS1 protein. However, further validation through *in vitro* studies (*Sadybekov & Katritch, 2023*) is required to confirm their efficacy.

## CONCLUSIONS

In conclusion, the limitations of current KFDV vaccines underscore the necessity for effective antivirals, emphasizing novel therapeutic approaches through *in silico* drug design. This study focused on targeting the NS1 protein, which is a multifunctional protein involved in viral replication, immune evasion, and host cell interactions. Computational methods, including molecular modeling, identified promising candidates: compounds L2, L3, and L5, alongside FDA-approved dasabuvir. These compounds demonstrated stability in 200 ns simulations, with significant MM-GBSA binding free-energy values: L2 ($-62.97 \pm 4.0$ kcal/mol), L3 ($-77.22 \pm 4.71$ kcal/mol), L5 ($-62.07 \pm 2.88$ kcal/mol), and dasabuvir ($-87.68 \pm 4.31$ kcal/mol). L2 and L3 showed comparable binding affinity to dasabuvir, suggesting their potential as effective NS1 protein-targeting drug candidates. Further validation through rigorous *in vitro* assays is crucial to confirm these findings and advance these candidates toward clinical application, offering new strategies for combating KFDV infection.

## ACKNOWLEDGEMENTS

The authors thank KLE Technological University, Hubballi, for extending research support and resources to carry out the present work. The author would like to express sincere gratitude to AlMaarefa University, Riyadh, Saudi Arabia, for providing the *in silico* software and analytical components of this research.

### Funding

The Deanship of Scientific Research at King of Khalid University provided funding for this work through Small Groups Project under grant number RGP2/426/46. This research was also funded by the Vision Group on Science and Technology, Karnataka, India, grant number GRD-996. One of the authors (Sharanappa Achappa) received financial support from Vision Group on Science and Technology, Karnataka, India (no. KSTePS/VGST/2020-21/RGS/F/GRD-996/54/2021-22/916). The funders had no role in

study design, data collection and analysis, decision to publish, or preparation of the manuscript.

## Grant Disclosures

The following grant information was disclosed by the authors:

King of Khalid University: RGP2/426/46.

Vision Group on Science and Technology, Karnataka, India: GRD-996, KSTePS/VGST/2020-21/RGS/F/GRD-996/54/2021-22/916.

## Competing Interests

The authors declare there are no competing interests.

## Author Contributions

- Sharanappa Achappa conceived and designed the experiments, performed the experiments, prepared figures and/or tables, authored or reviewed drafts of the article, and approved the final draft.
- Nayef Abdulaziz Aldabaan analyzed the data, prepared figures and/or tables, authored or reviewed drafts of the article, and approved the final draft.
- Ibrahim Ahmed Shaikh analyzed the data, prepared figures and/or tables, authored or reviewed drafts of the article, and approved the final draft.
- Mater H. Mahnashi analyzed the data, authored or reviewed drafts of the article, and approved the final draft.
- Shivalingsarj V. Desai conceived and designed the experiments, performed the experiments, authored or reviewed drafts of the article, and approved the final draft.
- Mufarreh Asmari analyzed the data, authored or reviewed drafts of the article, and approved the final draft.
- Mohammed Alasmary analyzed the data, authored or reviewed drafts of the article, and approved the final draft.
- Uday M. Muddapur conceived and designed the experiments, prepared figures and/or tables, authored or reviewed drafts of the article, and approved the final draft.
- Basheerahmed Abdulaziz Mannasaheb analyzed the data, authored or reviewed drafts of the article, and approved the final draft.
- Aejaz Abdullatif Khan analyzed the data, authored or reviewed drafts of the article, and approved the final draft.

## Data Availability

Code and raw data are available at Figshare:

Achappa, Sharanappa (2025). NS1 Simulation files. figshare. Software. https://doi.org/10.6084/m9.figshare.28737809.v1.

## Supplemental Information

Supplemental information for this article can be found online at http://dx.doi.org/10.7717/peerj.19954#supplemental-information.

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
