# Peer review of "Exploring phytochemicals as potential pharmacological inhibitors for NS1 protein of Kyasanur forest disease virus using virtual screening, molecular docking, and molecular simulation approach"

_PeerJ, doi:10.7717/peerj.19954_

## Round 0.1 · original submission · Major Revisions

I strongly recommend that you conduct a detailed line-by-line review of the manuscript. This will help identify areas where further clarity, precision, or corrections are needed. Attention to these details will significantly improve the overall quality of your work and demonstrate the seriousness of addressing the manuscript's flaws.

Reviewer 1 ·

Basic reporting

The manuscript entitled "Exploring phytochemicals as potential inhibitors for NS1 protein of Kyasanur Forest Disease Virus using virtual screening, molecular docking, and molecular simulation approach" by Achappa et al. employs a classical in silico approach to identify potential compounds targeting the KFDV NS1 protein. While the study holds potential, the manuscript can only be considered for publication after the authors address the raised comments and provide the necessary clarifications and improvements.

1. The introduction provides a good overview of KFDV geographical outbreaks, pathology, virus genomics, and the IMPPAT database. However, it lacks discussion on current advancements in drug design against KFDV, such as in silico studies of envelope binders or NS2B/NS3 inhibitors.
2. Too many unnecessary selfcitations.

Experimental design

1. SOPMA is used for secondary structure analysis, not 2D structure. Please specify the parameters used for analysis, such as similarity threshold and window width, and ensure corrections are made in the text and figure captions.
2. Please clarify the grid box size used in docking. The provided xyz coordinates mention 30 points, but the units are missing.
3. Why were simulations performed at 27°C, which is 10°C lower than the human body temperature? Please explain the rationale.
4. It is doubtful that the NS1 protein fits into a 10 Å × 10 Å × 10 Å periodic boundary box. Please verify this.
5. What ionic strength is assumed under "physiological conditions"? Please specify.
6. Why was a minimization run performed after a 12 ns NPT equilibration run (lines 241-243)? Please provide justification.
7. How many structures were used for MM-GBSA energy calculations? Please clarify.
8. How were the protonation states of amino acids determined prior to docking and MD simulations? Please explain the methodology.

Validity of the findings

1. Secondary structure analysis by SOPMA was conducted using secondary structure predictions. Could the authors compare these results with data from the generated 3D structure?
2. Please explain the validation graphs (Figures 2D and 2E). For example, Figure 2C shows 340 residues, whereas Figures 2D and 2E indicate 1,000 residues.
3. The axes in Figures 2E and 2F are difficult to read. Please improve their legibility.
4. Why was the Robetta model not refined, considering it has the lowest residue count in disallowed regions?
5. Include a paragraph discussing the known function of the NS1 protein, making clear connections to the selected binding pocket.
6. Why was the third docking campaign performed using AutoDock instead of AutoDock Vina, as in the first two campaigns?
7. The 2D structures in Table 7 are poorly visible. Please enhance their quality.
8. What were the final criteria for selecting the top 15 compounds? Were they based on docking scores? Specify the program used and explain why it was preferred.
9. Why is there no Vina score for dasabuvir?
10. Explain in the manuscript why dasabuvir was selected as a reference drug. It is a hepatitis C virus non-nucleoside NS5B polymerase inhibitor. Include a discussion on the known function of the KFDV NS1 protein in the explanation.
11. Use SI units consistently in graphs (e.g., "ns" instead of "nsec").
12. How was RMSD calculated? Relative to which geometry? Typically, RMSD is calculated relative to the first geometry, but here, it starts at 1.76 Å at t = 0 ns (line 483). Please clarify.
13. Use residue names and numbers (e.g., Ala101 or Gly100 to Trp150) in the text (e.g., line 491) and in figures (e.g., Figure 5B).
14. Line 516 mentions, "the ligand's RMSD points to a higher level of stability." Higher stability relative to what? Please clarify.
15. The average RMSD of 5.6 Å for the NS1-L3 complex is higher than typically accepted values for a "stable" conformation. Did the authors investigate the geometrical origin of the high RMSD values?
16. It is unusual for a small ligand (with ~10 times fewer atoms than the protein) to have an RMSD comparable to that of the protein. What happened to the ligand within the protein pocket? Did it undergo conformational changes? Did it move to other binding place?
17. Consider summarizing average RMSD and RMSF values in a table for better readability. The current textual statistics are hard to follow.
18. Figures 6 and S3 appear identical. Please confirm if this is intentional.
19. Clearly state which structure was used for the green bars in Figure 7 (ligand-protein interactions). Was it the final geometry from the MD simulations?
20. How did the radius of gyration and solvent-accessible surface area of the NS1 protein change during the dynamics?
21. Much emphasis is placed on analyzing interactions for docked structures and MD trajectories. Could the authors discuss similarities and differences between these two approaches?
22. It seems the same 2D interaction diagrams are presented for all complexes at 0 ns and 200 ns in Figure 10. Please confirm and correct if necessary.
23. Figure 12 lacks informativeness. Please include 3D structures to enhance clarity.
24. Consider selecting at least two ligand-protein distances and showing how they evolve during the simulations.
25. Why is the entropic contribution to the free energy of binding not included?
26. Include an overlap of 3D structures of the modeled KFDV protein and experimental structures of Zika or WNV NS1 proteins.

·

Basic reporting

The article Exploring phytochemicals as potential inhibitors for NS1 protein of Kyasanur Forest Disease Virus using virtual screening, molecular docking, and molecular simulation approach (#107504) is dedicated to the identification of potential antiviral compounds that inhibit the activity of the protein of KFDV. It is an important topic, and interesting results were obtained. However, the MS concentrated more on the description of protein modelling and molecular docking rather than on the discussion of the main aim of the study, which was the identification of candidate drugs and the possibilities of their potential application for the treatment of KFDV. Therefore, to make the MS interesting and useful for the readers, major revision is required.

Experimental design

The study is based on computational protein modeling, and all issues regarding the modeling, protein docking and molecular dynamic simulation were addressed after the first revision, according to the comments of the reviewer.

Validity of the findings

The major comment for the MS: the discussion requires major revision. The section fluctuates between a description of results (which should be in the Result section) and statements that similar methods were applied earlier for different viruses (an example is below).

776-781 Similar approach was carried by B.P.Joshi et al., 2022 for identification of novel phytochemical inhibitor of DLL3. Gurung et al., 2022 predicted potential inhibitors for corona virus from virtual screening of phytochemicals from IMPPAT database. Sha et al.,2024, Aljahdali et al.,2021, Roshniet al., 2022, Kalaria et al., 2020identified potential inhibitors from IMPPAT database for Nipah virus, Walleye dermal sarcoma virus, SARS-COV-2 and COVID-19 respectively.

This is an example of long paragraph validating the approach used in the study. In my opinion, it should be more concise, probably one sentence is enough, and it should be the Results section.

The Discussion session does not provide the reader with the discussion of the results obtained in this study. Thus, it is not clear how applicable citations like:
808According to the literature, ranmogenin A has shown inhibitory effects on human gastric tumor cells (Pan et al., 2003)
to the proposed used of identified compounds as antiviral agents.
Moreover, without mentioning in the discussion session, in Conclusion section, the authors state that:
852 Further validation through rigorous in vitro assays is crucial to confirm these findings and advance these candidates toward clinical application
As for the clinical application, it is known that several studies demonstrated high toxicity of solanogantamine, why is it not mentioned and discussed? Other compounds proposed as candidates in the study exhibit some toxicity, and again, they are not discussed.
Since performed research stated as the aim
157-158 provide foundational knowledge for the development of effective in silico-based antiviral therapies for KFDV.
The contribution of this study to the development of antiviral therapy as well as the potential limitations of identified candidates should be discussed.

Additional comments

In addition to the discussion, changes also should be made to the Introduction section.
Introduction
110-113 The KFDV genome encodes a polyprotein that undergoes proteolytic cleavage to generate various structural and non-structural proteins. Key proteins include the envelope glycoprotein E and several non-structural proteins (NS1, NS2A, NS2B,113 NS3, NS4A, NS4B, and NS5) (
In many studies, including recent comprehensive review of KFDV, it was pointed out that
The viral genome encodes a polyprotein consisting of three structural proteins (capsid-C, transmembrane-M, and envelope-E) and seven non-structural (NS) proteins (NS1, NS2a, NS2b, NS3, NS4a, NS4b, NS5) [13].
N S, Kandi V, G SR, Ca J, A H, As A, Kapil C, Palacholla PS. Kyasanur Forest Disease: A Comprehensive Review. Cureus. 2024 Jul 23;16(7):e65228. doi: 10.7759/cureus.65228. PMID: 39184677; PMCID: PMC11343324.
This should be at least mentioned in the introduction. Also, if the at least schematic structure of the virus is presented, it will definitely make easier reading and understanding of the MS.
The substantiation of NS1 as a target of the study is not quite clear. Thus, it is known that NS1 and NS5 protein-based subunit vaccines are being explored to develop and use against KFD.

Reviewer 3 ·

Basic reporting

No comment

Experimental design

No comments

Validity of the findings

No comments

Additional comments

The authors have provided valuable insights into an important public health concern by exploring the use of computational tools to identify therapeutic agents derived from phytochemicals for a significant tick-borne virus. Their work is particularly timely and relevant, given the growing impact of this virus in their region and its expanding reach.

Below are my comments

Line 105: check the spelling of the word replicating.

Line222 remove the word "computational" from parenthesis and incorporate it into the sentence; the phrase can read as "field of computational drug design
in line 754, change "a" to "an"
line 771  correct grammar "carried in"
779 add space

---

## Round 0.2 · Major Revisions

Kindly thoroughly revise your manuscript and carefully follow the reviewers’ comments, especially the careless formatting and reviewer 1's major comments

Reviewer 1 ·

Basic reporting

no comment

Experimental design

1) Line 276 – The authors define the number of points per dimension but do not specify the actual dimension of one point in terms of length units.
2) Line 288 – In their revision, the authors state that the KFDV NS1 protein fits within a 10×10×10 ų periodic box. However, a review of the Dengue NS1 protein reveals that the distance between nitrogen atoms at the N- and C-termini exceeds 45 Å. Please provide the structure of the protein and/or the complex within the simulation box for verification. Perhaps the authors intended to indicate a 10 Å water buffer rather than box dimensions?
3) Line 289 – The authors report conducting MD simulations at an ionic strength of 2.88 mM. Typically, simulations are carried out at 150 mM ionic strength, which is considered physiologically relevant.
4) The authors are encouraged to use image editing software such as Photoshop or GIMP to address earlier revision comments regarding SI units, axis labels, and residue naming. The figure conventions should align consistently with the manuscript text.

Validity of the findings

5) Since the authors confirmed that Figures 8 and S3 are identical, it is presumed that both replicates used the same input coordinates and velocities, which is incorrect. The authors are kindly requested to rerun the second replicate, maintaining the same initial geometry but generating velocities randomly (with a different seed) based on the Maxwell–Boltzmann distribution at the desired temperature.

Additional comments

6) Dasabuvir is a non-nucleoside inhibitor developed specifically for the NS5B RNA-dependent RNA polymerase of the hepatitis C virus. Although both NS5 and NS1 are nonstructural proteins in flaviviruses, they differ markedly in both function and structure. NS5 is an enzymatic protein involved in viral RNA replication, making it amenable to classical small-molecule inhibition. In contrast, NS1 is a multifunctional glycoprotein lacking enzymatic activity and primarily mediating immune evasion and viral pathogenesis. It lacks a defined active site, and its surface topology is structurally unrelated to that of NS5. As a result, Dasabuvir’s binding characteristics are not transferable to NS1, which requires a distinct design strategy focused on disrupting protein–protein interactions or modulating conformational states. Therefore, Dasabuvir is not an appropriate reference compound for NS1-targeted drug design.

·

Basic reporting

The article Exploring phytochemicals as potential inhibitors for NS1 protein of Kyasanur Forest Disease Virus using virtual screening, molecular docking, and molecular simulation approach is dedicated to the identification of potential antiviral compounds that inhibit the activity of the protein of KFDV. It is an important topic, and interesting results were obtained. The performed revision drastically improved the MS, making focus on actual goals of the study, and, in my opinion, it can be accepted for publication after minor revision. I recommend minor revision due to careless, inconsistent formatting. The MS should be carefully checked for different font types, fused words, etc. Below are just a few examples of problems with formatting:
293 Different font:
last 50 frames
868 three words are fused together:
11,530compoundsfrom
Through the whole discussion the name Solanum giganteum is written in smaller font that the rest of the text

Experimental design

Research questions were well defined and applicable methods were used.

Validity of the findings

Findings are valid and important.

Reviewer 3 ·

Basic reporting

No comment

Experimental design

No comments

Validity of the findings

No comments

Additional comments

The authors have adequately responded and incorporated my comments.

---

## Round 0.3 · accepted · Accept

Thanks for submitting the revised version. Now your article is suitable for publication.

·

Basic reporting

After the revision, the article meets the PeerJ criteria and should be accepted for publication in its current form.

Experimental design

Rigorous investigation performed to a high technical & ethical standard.

Validity of the findings

Data provided in the article are important, statistically sound, & controlled. Conclusions are well stated, linked to the original research question, & limited to supporting results.

Reviewer 3 ·

Basic reporting

No comment

Experimental design

No comment

Validity of the findings

No comment

Additional comments

I find the author’s response satisfactory and believe it adequately addresses the concerns previously raised.